# Endothelial cell clonal expansion in the development of cerebral cavernous malformations

Matteo Malinverno [1], Claudio Maderna[1], Abdallah Abu Taha[2], Monica Corada [1], Fabrizio Orsenigo [1], Mariaelena Valentino[1], Federica Pisati[1,3], Carmela Fusco[4], Paolo Graziano [5], Monica Giannotta[1], Qing Cissy Yu [6], Yi Arial Zeng [6], Maria Grazia Lampugnani [1,7], Peetra U. Magnusson [2] & Elisabetta Dejana [1,2,8]

Cerebral cavernous malformation (CCM) is a neurovascular familial or sporadic disease that is characterised by capillary-venous cavernomas, and is due to loss-of-function mutations to any one of three *CCM* genes. Familial CCM follows a two-hit mechanism similar to that of tumour suppressor genes, while in sporadic cavernomas only a small fraction of endothelial cells shows mutated *CCM* genes. We reported that in mouse models and in human patients, endothelial cells lining the lesions have different features from the surrounding endothelium, as they express mesenchymal/stem-cell markers. Here we show that cavernomas originate from clonal expansion of few *Ccm3*-null endothelial cells that express mesenchymal/stem-cell markers. These cells then attract surrounding wild-type endothelial cells, inducing them to express mesenchymal/stem-cell markers and to contribute to cavernoma growth. These characteristics of *Ccm3*-null cells are reminiscent of the tumour-initiating cells that are responsible for tumour growth. Our data support the concept that CCM has benign tumour characteristics.

[1] Vascular Biology Unit, The FIRC Institute of Molecular Oncology Foundation, Milan 20139, Italy. [2] Department of Immunology, Genetics and Pathology, Uppsala University, Uppsala 752 37, Sweden. [3] Histopathology Unit, Cogentech S.c.a.r.l, Milan 20139, Italy. [4] Division of Medical Genetics, Fondazione IRCCS-Casa Sollievo della Sofferenza, San Giovanni Rotondo, Foggia 71013, Italy. [5] Pathology Unit, Fondazione IRCCS-Casa Sollievo della Sofferenza, San Giovanni Rotondo, Foggia 71013, Italy. [6] The State Key Laboratory of Cell Biology, CAS Centre for Excellence in Molecular Cell Science, Institute of Biochemistry and Cell Biology, Shanghai Institutes for Biological Sciences, Chinese Academy of Sciences, 200031 Shanghai, China. [7] Mario Negri Institute for Pharmacological Research, Milan 20156, Italy. [8] Department of Oncology and Haemato-Oncology, School of Medicine, University of Milan, Milano 20122, Italy. Correspondence and requests for materials should be addressed to M.M. (email: matteo.malinverno@ifom.eu) or to E.D. (email: elisabetta.dejana@ifom.eu)

Cerebral cavernous malformation (CCM) is a vascular disease that is characterised by capillary–venous cavernomas. These malformations are almost exclusively located in the central nervous system and they can cause micro bleeds that can lead to epileptic seizures and cerebral haemorrhage[1–5]. CCM occurs in familial and sporadic forms. Familial cases follow autosomal-dominant inheritance due to loss-of-function mutations in one of the three genes known as *CCM1/KRIT1*, *CCM2/malcavernin* or *CCM3/PDCD10*. The inherited forms of CCM develop multiple vascular lesions that tend to increase in number and size with age. In contrast, the sporadic form of CCM is characterised by only a single lesion. This difference in the lesion burden between familial and sporadic CCM suggests that CCM pathogenesis might follow a two-hit, biallelic molecular mechanism that is similar to that seen for tumour-suppressor genes. Moreover, a remarkable feature of familial CCM is that, although all of the endothelial cells have a heterozygote loss-of-function mutation to one of the three *CCM* genes, the malformations are only found in a few localised regions of the brain microcirculation. Furthermore, it has been shown that, for human sporadic cavernomas, only a small fraction of endothelial cells have a null mutation for the *CCM* genes[6–9]. Considering that the double hit is a rare event, this suggests that a small number of mutated endothelial cells appear to be enough to trigger the malformations.

In our previous studies, we reported that in mouse models of CCM and in human patients the endothelial cells lining cavernomas have different features than the surrounding endothelial cells of the same vessel. Specifically, the endothelial cells in the lesions show a mixed phenotype that combines both endothelial and mesenchymal features in a way similar to endothelial cells that are undergoing endothelial-to-mesenchymal transition (EndMT). Most importantly, these cells also express a relatively large set of stem cell markers (e.g., *Cd44*, *Id1*, *Slug*, *Klf4*, *Sca1*)[10–14]. These characteristics are reminiscent of the so-called tumour-initiating cells or tumour stem cells that are responsible for tumour growth and metastatic differentiation[15–17].

We show here that cavernomas originate from clonal expansion of a small number of $Ccm3^{-/-}$ endothelial cells that express EndMT and progenitor markers. These cavernoma-initiating cells can attract the surrounding wild-type endothelial cells and induce their mesenchymal transition, coinciding with a strong and time-dependent increase in the size of the cavernomas. Our findings support the concept that CCMs have benign tumour characteristics and are consistent with the observations of mosaicism in human cavernomas. Furthermore, data indicate that a minimal proportion of *Ccm3*-null endothelial cells can induce large-sized malformations, as in the human disease[8,9]. This concept is also in agreement with the fact that *Ccm3* is a tumour suppressor[18,19] and its deletion may be correlated to benign brain tumours[20].

## Results

**Cavernomas have clonal origin.** To follow the clonal expansion of endothelial cells, we took advantage of the *Cdh5(PAC)*-Cre-ER[T2]/R26R-*Confetti* mouse that carries the stochastic and multicolour reporter Brainbow2.1 in the R26 locus (R26R-*Confetti*[21,22]) and the inducible Cre-ER[T2] recombinase that is driven by the endothelial-specific promoter of VE-cadherin (*Cdh5(PAC)*-Cre-ER[T2])[23]. After a single injection of tamoxifen, this Cre-mediated recombination promotes endothelial cells to randomly express any one of the four fluorescent proteins to form a mosaic (Fig. 1a, b). To study whether the CCM lesions have a clonal origin, the *Cdh5(PAC)*-Cre-ER[T2]/R26R-*Confetti* mice were crossed with *Ccm3*[f/f] mice (*Cdh5(PAC)*-Cre-ER[T2]/*Ccm3*[f/f]/R26R-*Confetti*), which develop

CCM lesions upon tamoxifen injection that resemble the human pathology[24].

Injection of tamoxifen on postnatal day 1 (P1) induced deletion of the *Ccm3* gene and expression of one of the four fluorescent proteins in an endothelium-specific manner. By P8, the retina showed vascular malformation at the front, with large areas of clonal expansion (Fig. 1a). In the cerebellum, where most of the cavernomas were formed in this model (Fig. 1b, f), the majority of the small lesions appeared to be composed of cells of the same colour, which thus suggested their clonal origin. Larger lesions had a more complex composition, with clonal areas surrounded by regions with endothelial cells of mixed colours (Fig. 1b–f and Supplementary Movies 1–6). This suggested that, after the first clonal growth, the adjacent lesions might fuse or that surrounding cells might be recruited into the lesion.

The clonal expansion presupposes an increased cell proliferation of $Ccm3^{-/-}$ cells within the lesions, as we have previously shown in vivo[24]. *Ccm3* is known to have a pivotal role in regulating cell survival and cell death, and anti-apoptotic[25–27] as well as pro-apoptotic[28–31] functions have been reported in different cell types. Nevertheless, whether the increase in cell proliferation of endothelial cells lining the cavernomas is directly dependent on loss of *Ccm3* is not completely understood. Here we show that the loss of *Ccm3* is sufficient to increase the proliferation rate of endothelial cells and to drive the entrance into the S-phase, while the re-expression of the gene decreased cell proliferation to wild-type level (see Supplementary Figs. 1, 2, 13 and 14 for more details).

In parallel, we have tested the activated caspase 3 protein levels in both $Ccm3^{+/+}$ and $Ccm3^{-/-}$ conditions and we have not seen any significant difference, arguing that the deletion of *Ccm3* could not be sufficient to inhibit the endothelial cell apoptosis under physiological conditions.

**Large cavernomas are mosaics.** This 'fast progression' acute mouse model of $Ccm3^{-/-}$ CCM is lethal in around 12 days, and it does not allow detailed analysis of the temporal steps of cavernoma formation. To overcome this limitation, we generated a slow progression model (i.e. chronic) of $Ccm3^{-/-}$ CCM by giving a low dose of tamoxifen to these *Cdh5(PAC)*-Cre-ER[T2]/*Ccm3*[f/f] mice (Fig. 2 and Supplementary Movie 7), and we analysed them at different developmental stages (i.e. P8, P14 and P30). Under this condition, the mice reached adulthood, and as in the human CCM disease, only a small proportion of the cells (<15%) showed *Ccm3* deletion (Supplementary Figs. 3a, 11 and 12).

In the acute model, the lesions were preferentially located in the cerebellum and retina, while in the chronic model they were also in other brain regions, including the hippocampus and the olfactory bulb, similar to the human disease[32,33] (Fig. 2e).

While the number of lesions remained substantially constant over time, at P14 and P30 the mean area, and thus the total lesioned area, strongly increased (Fig. 2f–h).

Taking all of these characteristics into account, the chronic model appears to be more suitable for studies on the progression of human cavernomas.

In the chronic model of the *Cdh5(PAC)*-Cre-ER[T2]/*Ccm3*[f/f]/R26R-*Confetti* mice, not all the endothelial cells lining the large cavernomas were *Confetti* positive; i.e. $Ccm3^{-/-}$. In contrast, most of the endothelial cells were *Confetti* negative, and therefore they had not undergone *Ccm3* recombination (Fig. 3a–d and Supplementary Movie 8).

The stochastic nature of the recombination in the *Confetti* system cannot exclude that, by chance, two adjacent cells express the same colour without deriving by the same cell. In order to assess the frequency of this event, and to better quantify the

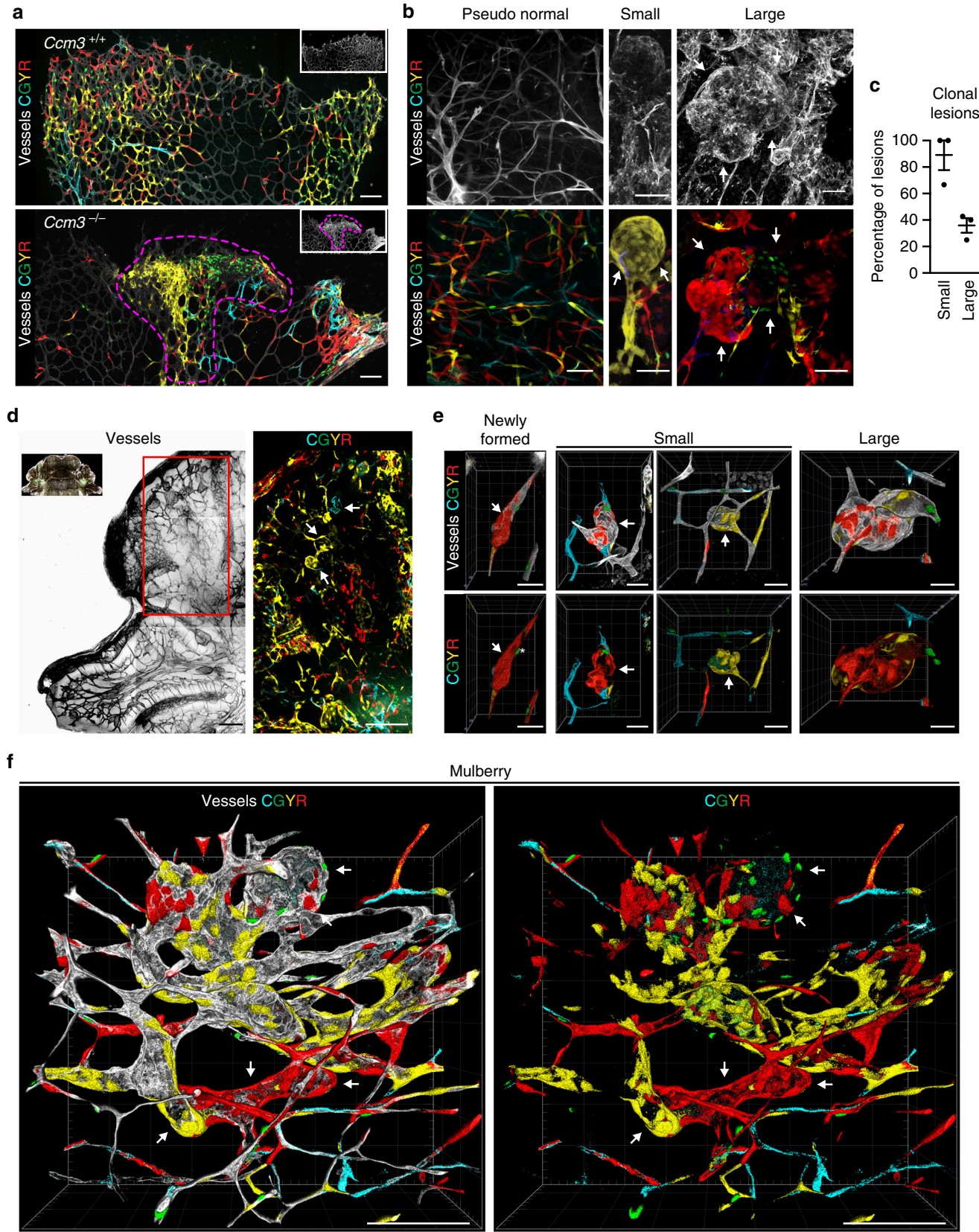

clonal expansion within the lesions, we performed a more quantitative analysis on cerebellum from *Cdh5(PAC)*-Cre-ER$^{T2}$/ *Ccm3*$^{f/f}$/R26R-*Confetti* mice and their *Ccm3*$^{+/+}$ counterpart after injection of low tamoxifen at P2 (Fig. 3e–g, Supplementary Fig. 4). We considered a clone any group of contiguous cells

expressing the same colour within a vessel and counted the number of cells that composed the clone. In the vessels of *Ccm3*$^{+/+}$ mice, the average size of clones was $1.14 \pm 0.016$ cells, with 88% of single cells and no clone >4 cells. Likewise, in normal vessels of the *Ccm3*$^{-/-}$, the average size of clones was $1.19 \pm 0.014$

**Fig. 1** Cavernomas have clonal origins. *Cdh5(PAC)*-Cre-ER$^{T2}$/R26R-*Confetti* or *Cdh5(PAC)*-Cre-ER$^{T2}$/*Ccm3*$^{fl/fl}$/R26R-*Confetti* mice following tamoxifen induction of the four fluorescent proteins and of *Ccm3* deletion at 1 day after birth, with analysis at day 8. **a** Representative images of vessels from retinas of *Ccm3*$^{+/+}$ (upper panel) or *Ccm3*$^{-/-}$ (lower panel) mice stained for Isolectin B4. *Ccm3*$^{+/+}$ retina showed random expression of the four fluorophores to form a mosaic; the retina vasculature of *Ccm3*$^{-/-}$ mice showed cerebral cavernous malformations and clonal regions composed of cells of the same colour. **b** Representative images of cerebellum from *Ccm3*$^{-/-}$ mice with vessels stained for Podocalyxin (left panel) or Isolectin B4 (central and right panels). Small cavernomas are mostly formed by cells of the same colour, while larger cavernomas show a mixed composition. **c** Quantification of clonality of the lesions, divided according to small and large lesions on the basis of the longer diameter (cut-off, 60 µm). Lesions were considered clonal when composed of ≥80% cells of the same colour. Each dot represents the percentage of clonal lesions in an animal; data are means ± SE. Source data are provided as a Source Data file. **d** Representative tiling of the cerebellum from P8 *Ccm3*$^{-/-}$ mice with vessels stained for Podocalyxin. **e, f** Three-dimensional (3D) reconstruction of representative 'newly formed', 'small', 'large' and 'mulberry' lesions. CGYR stands for CFP, GFP, YFP and RFP. Scale bars, 100 µm (**a**, **d**, **f**); 50 µm (**b**); 40 µm (**e**); white arrows point at lesions. See also Supplementary Movies 1–6 for animated 3D reconstruction of these lesions

cells, with 85% of single cells and no clone >4 cells. Within the lesions instead, the clones showed an average size of 9.35 ± 0.867 cells, with 43% of clones >4 cells. Moreover, around 12% of clones had a size from 10 up to 60 cells. A small fraction of clones of more than one cell (i.e. two or three cells) within normal vessels could be explained by either cell division (which can be expected in a developing vasculature) or the chance that two adjacent cells express the same colour. The presence of clones of significantly larger size, i.e. ten or more cells, however, strongly support the concept that an unusual expansion has occurred.

The quantification also showed that there is no different distribution of the four fluorophores between lesions and normal vessels and that the CFP is slightly under represented among the four fluorescent proteins (Fig. 3e).

While this paper was under revision, others reported on clonal expansion of endothelial cells under pathological conditions using different mathematical approaches[34,35].

Fluorescence-activated cell sorting (FACS) analysis of endothelial cells isolated from the *Cdh5(PAC)*-Cre-ER$^{T2}$/*Ccm3*$^{f/f}$/R26R-*Confetti* mice treated with low tamoxifen showed that only 11.0 ± 7.4% endothelial cells underwent recombination of *Ccm3* (Supplementary Fig. 3a). In addition, quantitative real-time polymerase chain reaction (RT–qPCR) confirmed that endothelial cells positive for *Confetti* also showed recombination for the *Ccm3* gene, while the *Confetti*-negative cells expressed the same level of *Ccm3* transcript as the wild-type endothelial cells (Supplementary Fig. 3b). Overall, these data support the concept that, after the initial clonal expansion of a few endothelial cells that have undergone recombination, further development of the cavernomas was promoted by recruiting *Ccm3*$^{+/+}$ cells that had not undergone recombination (i.e. wild-type endothelial cells).

In the acute model, a large proportion of the endothelial cells (>80%) had biallelic inactivation of one of the *Ccm* genes, but only the cells that lined the cavernomas showed EndMT markers[24,36]. Double immunostaining for endothelial (ERG1, COLLAGEN IV, PECAM1) and EndMT markers (i.e., FSP1, pSMAD3, ID1, FN1, SCA1, αSMA, KLF4) in the chronic model showed that at early stage (P8) small lesions are composed mostly by endothelial cells that expressed higher levels of EndMT markers if compared to cells of surrounding normal vessels, while at later stages (P14), in larger cavernomas only a subset of endothelial cells underwent EndMT (Fig. 4).

At P8, almost 100% of the endothelial cells that lined the cavernomas were KLF4-positive (i.e. bona fide endothelial cells undergoing EndMT), which then declined during the increase in the size of the cavernomas to 70% KLF4-positive cells at P14 and 40% KLF4-positive cells at P30 (Fig. 4b).

To further confirm this data, we crossed *Cdh5(PAC)*-Cre-ER$^{T2}$/ *Ccm3*$^{f/f}$ mice with a different reporter mouse, for R26-*EYFP*, which is widely used for cell tracking[37]. With this reporter, mosaicism of the recombination of the endothelial cells lining the lesions was confirmed, with a progressive increase of

EYFP-negative, i.e. *Ccm3*$^{+/+}$, endothelial cells being recruited into the large cavernomas (Fig. 5a). Moreover, co-immunostaining demonstrated that the cells lining the cavernomas and undergoing EndMT transition had an endothelial origin (Fig. 5b–f), as they also expressed the EYFP. The wild-type endothelial cells that were recruited into the lesions also overexpressed EndMT markers, including KLF4, ID1, pSMAD3, SCA1 and αSMA (Fig. 5b–f).

**$Ccm3^{-/-}$ cells recruit $Ccm3^{+/+}$ endothelial cells.** To further prove that *Ccm3*$^{-/-}$ cells can form cavernomas and recruit surrounding *Ccm3*$^{+/+}$ cells, we isolated endothelial cells from acute P7 *Cdh5(PAC)*-Cre-ER$^{T2}$/*Ccm3*$^{f/f}$/R26R-*Confetti* mice and we injected these cells into the brains of adult wild-type mice. After 4 days, *Ccm3*$^{-/-}$ Confetti-positive cells formed abnormal vessels that comprised also Confetti-negative cells (Fig. 6a, b). In order to confirm the recruitment of *Ccm3*$^{+/+}$ cells by injecting *Ccm3*$^{-/-}$ cells, we isolated endothelial cells from acute P7 *Cdh5(PAC)*-Cre-ER$^{T2}$/*Ccm3*$^{f/f}$, Confetti negative mice and injected them in *Cdh5 (PAC)*-Cre-ER$^{T2}$/R26R-*Confetti* adult mice. Here also, *Ccm3*$^{-/-}$ cells generated abnormal vessels that recruited Confetti-positive, thus *Ccm3*$^{+/+}$, endothelial cells of the host (Fig. 6c, d) In parallel, we engrafted the same *Ccm3*$^{-/-}$ cells in a Matrigel plug subcutaneously into *Cdh5(PAC)*-Cre-ER$^{T2}$/R26R-*Confetti* mice. After 10 days, the injected *Ccm3*$^{-/-}$ endothelial cells formed abnormal vessels and recruited Confetti-positive endothelial cells coming from the host (Fig. 6e, f). Overall these data confirmed that *Ccm3*$^{-/-}$ endothelial cells were able to form CCM lesions and to recruit normal surrounding cells when implanted in a wild-type context.

To investigate the mechanism(s) that regulates the cross-talk between these *Ccm3*$^{-/-}$ and *Ccm3*$^{+/+}$ cells in greater depth, we developed a set of in vitro assays. First, we co-cultured *Ccm3*$^{+/+}$ and *Ccm3*$^{-/-}$ immortalised endothelial cells. When cultured alone, both the *Ccm3*$^{+/+}$ and *Ccm3*$^{-/-}$ cells formed a typical endothelial monolayer (Supplementary Fig. 5c, d). However, when the *Ccm3*$^{-/-}$ cells were mixed with the *Ccm3*$^{+/+}$ cells they progressively formed spheroids while the *Ccm3*$^{+/+}$ endothelial cells still grew in a monolayer (Fig. 7a, b; Supplementary Fig. 6). These spheroids were composed of 26.8 ± 3.19% *Ccm3*$^{+/+}$ cells. In the areas of close contact, the *Ccm3*$^{+/+}$ cells became integrated into the *Ccm3*$^{-/-}$ spheroids. Time-lapse movies confirmed that the wild-type (i.e. *Ccm3*$^{+/+}$) cells were actively recruited into the *Ccm3*$^{-/-}$ spheroids, rather than just being entrapped through physical interactions (Supplementary Movie 9). Moreover, when the established spheroids were detached and placed onto a new *Ccm3*$^{+/+}$ cell monolayer, these new wild-type cells were actively recruited and entered the spheroids (Fig. 7c–e; Supplementary Fig. 7). Time-lapse movies of wound-healing experiments showed the different migration behaviours of these *Ccm3*$^{+/+}$ and *Ccm3*$^{-/-}$ cells (Supplementary Fig. 8a, b, and Supplementary Movie 10). These also showed that, as soon as the

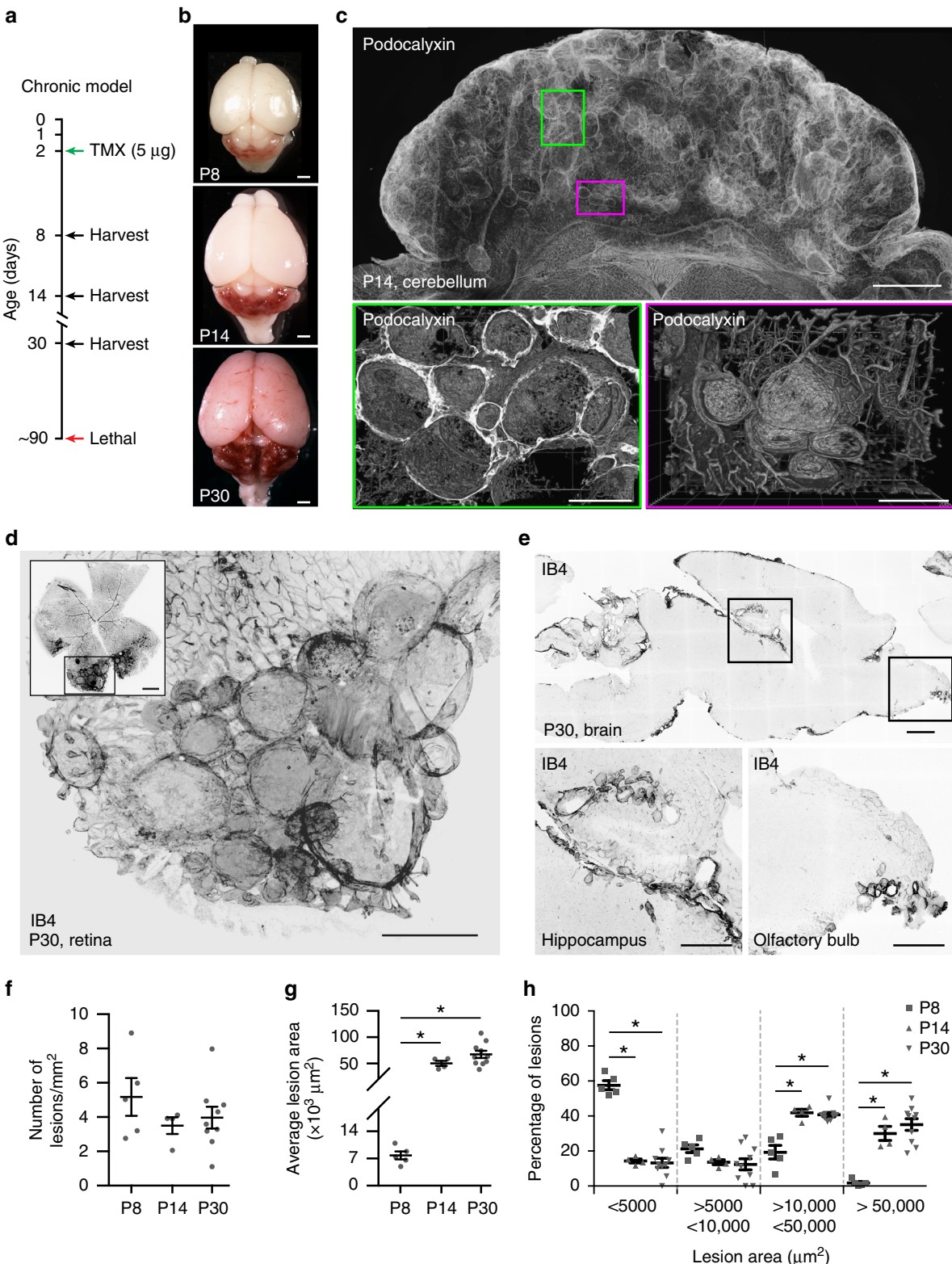

$Ccm3^{+/+}$ cells came into contact with the $Ccm3^{-/-}$ cells, they acquired mesenchymal features, with filopodia that protruded into the $Ccm3^{-/-}$ monolayer (Supplementary Fig. 8c–d, and Supplementary Movies 10 and 11).

**$Ccm3^{-/-}$ cells induce EndMT in $Ccm3^{+/+}$ endothelial cells.** As well as being recruited by the $Ccm3^{-/-}$ cells, the $Ccm3^{+/+}$ cells

were stimulated to undergo EndMT. The $Ccm3^{+/+}$ cells that were isolated from these spheroids expressed higher levels of the EndMT markers *Fsp1*, *Cd44* and *Fn1* and decreased levels of the endothelial-specific *Claudin5*, as compared to non-recruited $Ccm3^{+/+}$ cells that remained as monolayer at the bottom (Fig. 7g). To exclude that the upregulation of EndMT markers was merely due to the fact that they grew in three-dimensional

**Fig. 2** The slow progression model of cerebral cavernous malformation (CCM) develops large lesions. A chronic model of CCM was generated by treating *Cdh5(PAC)*-Cre-ER^T2/*Ccm3*^f/f/*R26R-Confetti* mice with low-dose tamoxifen. **a** Scheme of treatment with tamoxifen at P2 and analysis at P8, P14 and P30. **b** Representative photographs of whole brains from chronic P8, P14 and P30 mice; scale bar: 100 μm. **c** Representative tiling of a cerebellum at P14 showing the distribution of lesions; upper panel shows a projection from a 1-mm-thick section; lower panels show three-dimensional reconstruction of corresponding regions. Lower left panel was rotated by 90°; vessels were stained for Podocalyxin; scale bars: 1000 μm lower magnification, 300 μm higher magnification. **d** Representative confocal image of P30 retina stained for Isolectin B4 (black vessels) showing large cavernomas at the front; scale bar: 500 μm lower magnification, 100 μm higher magnification. **e** Representative confocal images of P30 brain stained for Isolectin B4 (black vessels) showing cavernomas in the hippocampus and the olfactory bulb; 1000 μm lower magnification, 500 μm higher magnification. **f, g** Quantification of total number and mean area of lesions in chronic P8, P14 and P30 mice. Data are means ± SE; *p* < 0.001 among groups (one-way analysis of variance (ANOVA)); *$p$ < 0.01 (Tukey's post hoc test). **h** Lesions were divided into four groups according to their size, as shown. Data are means ± SE. *$p$ < 0.001 among groups (one-way ANOVA); *p* < 0.01 (Tukey's post hoc test); a total of 1698 lesions have been counted from 5 animals at P7, 4 animals at P14 and 9 at P30. Source data are provided as a Source Data file

(3D), we cultured *Ccm3*^+/+ alone as spheroids on a methylcellulose matrix and compared their gene expression to the same cells cultured as monolayer (Supplementary Fig. 9). After 7 days, the cells cultured as spheroids showed no increase of *Fn1* and *Claudin5* and a slight increase of *Cd44*, which was not comparable to the one induced by the co-culture with *Ccm3*^−/− cells. Only *Fsp1* showed a significant upregulation under 3D conditions. Therefore, we can assess that the molecular modifications of recruited *Ccm3*^+/+ cells are mostly induced by the presence of *Ccm3*^−/− cells rather than by the 3D culture itself. Moreover, the induction of EndMT by the co-culture was reversible, as once the *Ccm3*^+/+ cells isolated from the spheroids were put back in culture alone, they reverted to the expression level of the *Ccm3*^+/+ cells cultured as monolayer (Supplementary Fig. 9).

We also asked whether the EndMT phenotype induced by the deletion of *Ccm3* could be rescued by re-expression of the gene. To this aim, we infected *Ccm3*^−/− cells with a lentiviral vector that expresses the human *CCM3* gene fused to EGFP (Supplementary Fig. 1), to generate a reconstituted cell line (*Ccm3*^−/−-hCCM3-EGFP). Interestingly, the *Ccm3*^−/−-hCCM3-EGFP cells reverted to wild-type phenotype in terms of proliferation rate (Supplementary Fig. 1), morphology, adherens junctions and expression of EndMT markers (Supplementary Fig. 1).

Furthermore, conditioned medium from these *Ccm3*^−/− cells increased the migration rates of *Ccm3*^+/+ cells (Fig. 7f) and induced them to upregulate their EndMT markers (Fig. 7h), compared to the negative control of conditioned medium from *Ccm3*^+/+ cells.

Taken together, these data support the concept that *Ccm3*^−/− cells can attract *Ccm3*^+/+ cells and induce them to undergo EndMT, as it has also been reported in vivo. The data on the effects of this conditioned medium here (Fig. 7f) suggested that secreted soluble factors are involved in the *Ccm3*^−/−-cell-mediated recruitment of the *Ccm3*^+/+ endothelial cells in cooperation with direct interaction of *Ccm3*^−/− and *Ccm3*^+/+ cells.

We reported previously that *Ccm1*^−/− cells produce high amounts of BMP6 and show significant phosphorylation of SMAD2/3 and SMAD1/5[36,38] also under resting conditions. Furthermore, stimulation of wild-type endothelial cells with BMP6 induced upregulation of their EndMT markers, which was inhibited by the BMP pathway general inhibitor DMH1[36,39]. Last but not least, BMP6 is a strong inducer of endothelial chemotaxis[40,41].

Here also, these *Ccm3*^−/− cells expressed high levels of BMP6 and showed increased phosphorylation of SMAD1 and SMAD3 (Supplementary Fig. 10a). The treatment of the *Ccm3*^+/+ cells with BMP6 promoted upregulation of the EndMT markers *Klf4*, *Sca1*, *Fsp1*, *Cd44* and *Id1* (Supplementary Fig. 10d), which was blocked by bone morphogenetic protein (BMP) inhibition. Moreover, BMP6 increased the migration rates of wild-type (i.e. *Ccm3*^+/+) cells and the BMP inhibitors blocked this effect

(Supplementary Fig. 10e). Consistent with this, in a wound-healing assay, treatment with the BMP inhibitor DMH1 blocked the increased migration rate induced by *Ccm3*^−/−-cell-conditioned medium (Fig. 7f).

Overall, these data are suggestive of a role of BMP6, possibly in combination with other inflammatory cytokines, in the attraction of wild-type endothelial cells in the cavernomas by *Ccm3*^−/− cells.

**Endothelial progenitors are involved in cavernoma formation.** We have shown that cavernomas originate from clonal expansion of *Ccm3*^−/− endothelial cells that co-express endothelial and mesenchymal/stem cell markers and trigger the recruitment of *Ccm3*^+/+ endothelial cells. Recently, vessel-resident endothelial progenitors have been identified and characterised on the basis of the expression of different markers[10,12,42,43], which included *Cd157*/Cd200[44], *Procr*[45] and *Peg3/PW1*[46]. These progenitors co-expressed endothelial and mesenchymal/stem cell markers and underwent clonal expansion to generate new blood vessels[34].

Taking into consideration the strong similarities with other tissue stem cells, it is tempting to speculate that the early steps of cavernoma formation are due to the clonal expansion of resident endothelial progenitors. This is supported by the endothelial cells in the early lesions, which express most of the stem/progenitor markers, including *Id1*, *Sca1*, *Cd44* and *Klf4*, and more interestingly, also the specific endothelial progenitor marker *Protein C Receptor* (*Procr*; Fig. 8a, b). Furthermore, it has been shown that the number of endothelial progenitors declines with the age of these mice, as higher numbers in neonates and almost undetectable levels in adults,[46] which is consistent with the decline in the number of CCM malformations that form when *Ccm3* is deleted at later time points after birth[24,47]. Here we confirmed by FACS analysis that *Procr* is expressed in a small population, 4.4 ± 1.5%, of brain endothelial cells in wild-type mice at P2 (Fig. 8d), which is consistent with what is already reported in other tissues[45]. Moreover, RNAseq analysis made on endothelial cells from the wild-type brains[48] showed that the expression of *Procr* decreases over time after birth (Fig. 8e).

To further test this hypothesis, we asked whether inactivation of *Ccm3* in progenitor cells would specifically induce their clonal expansion and early steps of CCM development. To this end, we crossed the *Ccm3*^f/f mice with the *Procr*^CreERT2-IRES-tdTomato/+ knock-in mouse model, in which a CreERT2-IRES-tdTomato cassette is driven by the promoter of *Procr*[45,49]. The *Procr*^CreERT2-IRES-tdTomato/+/*Ccm3*^f/f mouse model allowed induction of the tamoxifen-dependent *Ccm3* deletion specifically in *Procr*^+ endothelial progenitors.

After recombination, the mice developed lesions both in the retina and the cerebellum (Fig. 8c), in a way that recapitulates the formation of lesions in the chronic model. By crossing the *Procr*^CreERT2-IRES-tdTomato/+/*Ccm3*^f/f mouse model with the

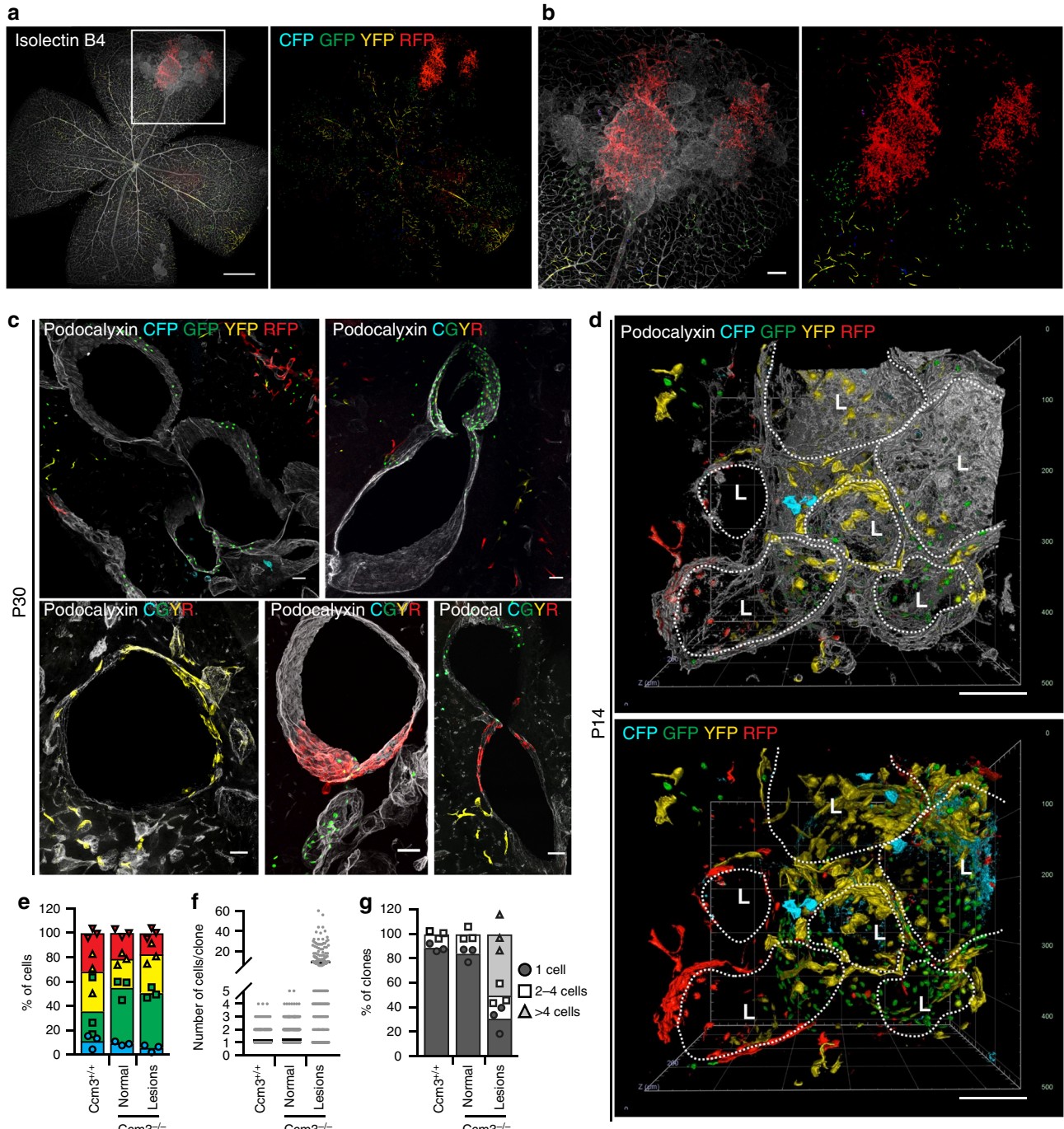

**Fig. 3** Cavernomas are composed of clones of $Ccm3^{-/-}$ cells that attract $Ccm3^{+/+}$ cells. Representative images of retina (**a**, **b**) and cerebellum (**c**) from chronic $Cdh5(PAC)$-Cre-ER$^{T2}$/$Ccm3^{f/f}$/R26R-*Confetti* mice at P30. The vessels are counterstained with Isolectin B4 (**a**, **b**) or Podocalyxin (**c**) and are displayed as greyscale. **d** Three-dimensional reconstruction of mulberry cavernomas from P14 brain. The vessels are counterstained with Podocalyxin; L stands for Lumen. Scale bars, 500 μm (**a**); 100 μm (**b**); 50 μm (**c**); 100 μm (**d**). **e**–**g** Analysis of clones in normal vessels of $Cdh5(PAC)$-Cre-ER$^{T2}$/R26R-*Confetti* mice and normal vessels and lesions of $Cdh5(PAC)$-Cre-ER$^{T2}$/$Ccm3^{f/f}$/R26R-*Confetti* mice. **e** Distribution of the four fluorophores among clones. Data are means ± SE; $n = 3$ mice in each group. **f** Quantification of the number of cells that compose each clone. Data are means ± SE; $n = 3$ mice in each group. **g** Distribution of clone size. Clones were divided according to the number of cells they are composed of; each dot represents an animal. A total of 2371 clones were analysed from 3 mice in each group. Source data are provided as a Source Data file

R26-*EYFP* reporter, we confirmed that the lesions were composed of EYFP$^+$ cells, thus originating from *Procr*$^+$ endothelial progenitors (Fig. 8f). Finally, the endothelial cells lining the cavernomas overexpressed EndMT markers if compared to the cells of surrounding normal vessels (Fig. 8g–j),

thus resembling the pathogenesis already shown by the *Cdh5 (PAC)*-Cre-ER$^{T2}$/$Ccm3^{f/f}$ model.

These data thus support the idea that endothelial progenitors are responsible for the triggering of cavernoma formation upon *Ccm3* deletion.

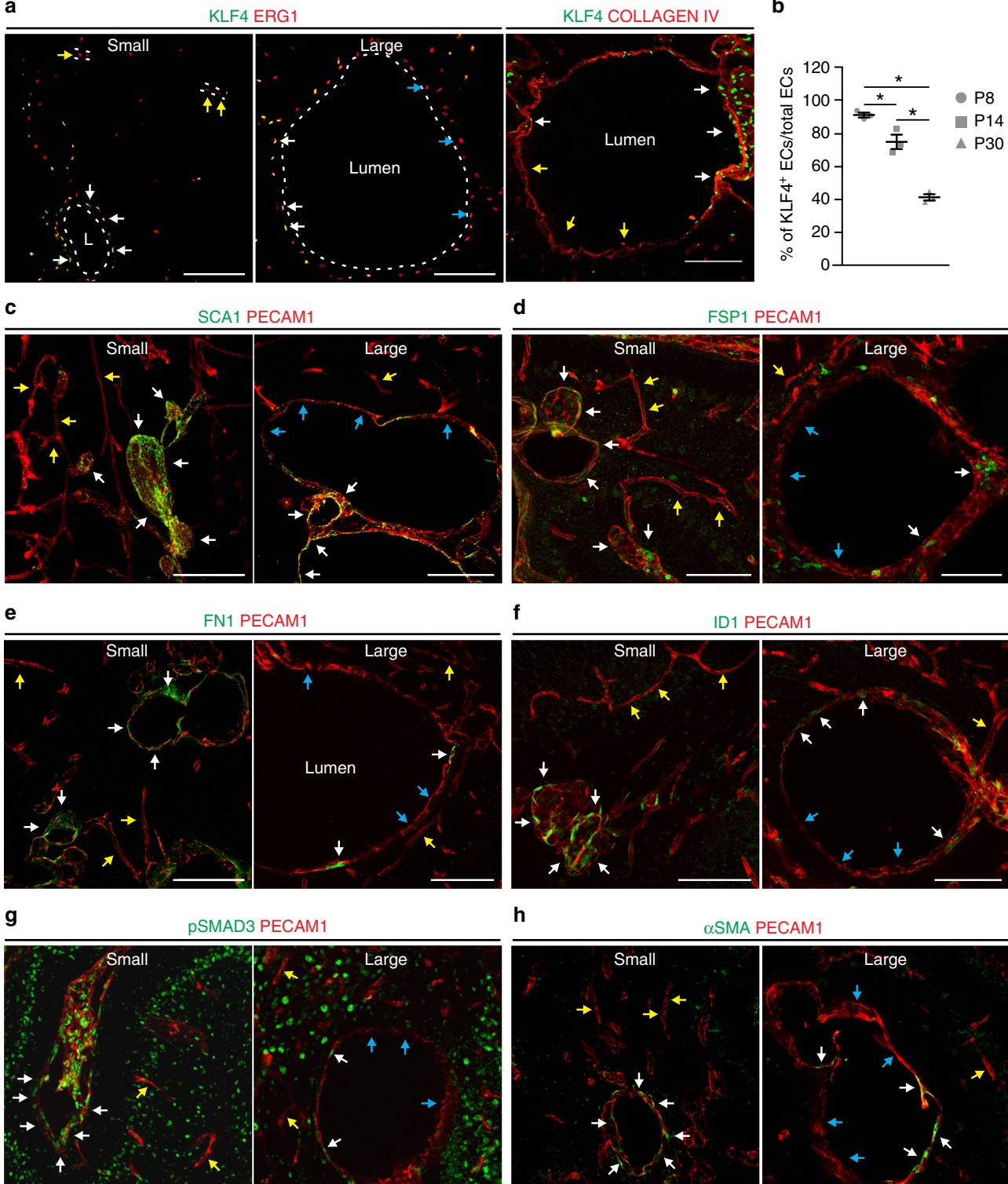

**Fig. 4** Large cavernomas are mosaic. **a** Representative images of chronic *Cdh5(PAC)*-Cre-ER[T2]/*Ccm3*[f/f] mice co-stained for KLF4 (green) and either ERG1 or COLLAGEN IV (red). **b** Quantification of the percentage of KLF4-positive endothelial cells lining the cavernomas of chronic P8, P14 and P30 mice. Data are means ± SE; $p < 0.001$ among groups (one-way analysis of variance); *$p < 0.01$ (Tukey's post hoc test); $n = 3$ mice in each group. Source data are provided as a Source Data file. **c–h** Representative images of brain sections co-stained for PECAM1 (red) and mesenchymal markers (green), showing small lesions from P8 mice and large cavernomas from P14 mice. White arrows, endothelial cells positive for mesenchymal markers within the lesions; yellow arrows, endothelial cells negative for mesenchymal markers in normal vessels; blue arrows, endothelial cells negative for mesenchymal markers within the lesions. Scale bars, 100 μm

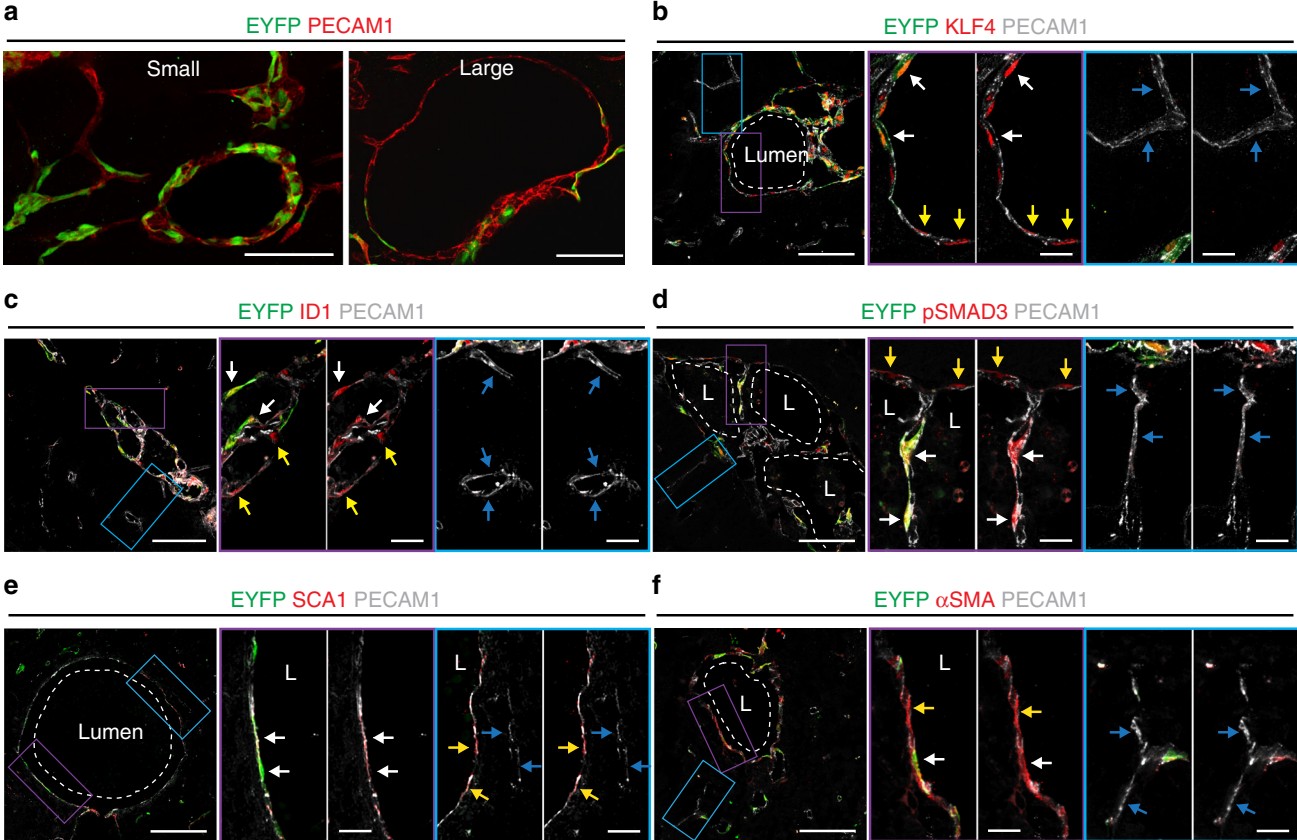

**Fig. 5** $Ccm3^{+/+}$ cells recruited into cavernomas undergo phenotypical modifications. **a** Representative images from chronic $Cdh5(PAC)$-Cre-ER$^{T2}$/$Ccm3^{f/f}$/ R26-EYFP mice showing small lesion from P8 mice and large cavernoma from P14 mice cerebellum. **b**–**f** Brain sections immunostained for EYFP (green), PECAM1 (grey) and the mesenchymal markers KLF4 (**b**), ID1 (**c**), pSMAD3 (**d**), SCA1 (**e**) and αSMA (red) (**f**). White arrows, endothelial cells positive for EYFP in the cavernomas; yellow arrows, endothelial cells with no recombination in the cavernomas; blue arrows, endothelial cells in normal vessels. Scale bars, 100 μm

## Discussion

Taken together, our data explain some of the crucial aspects of the development and evolution of CCM, while also opening novel therapeutic opportunities. Using a slowly evolving $Ccm3^{-/-}$ model system, we have demonstrated that, upon induction of homozygote $Ccm3$ mutation, only a few of the endothelial cells undergo clonal expansion and attract $Ccm3^{+/+}$ cells, which, in turn, contribute to the growth of the malformations. This is in agreement with the human disease, where following a double-hit process[6–9], a few heterozygote endothelial cells acquire homozygosity for the $Ccm$ gene inactivation and trigger the formation of cavernomas. Consistently in both mice and humans, the endothelial cells lining the lesion appear as a mosaic, i.e. formed by both mutated and wild-type cells. Therefore, an important event in the evolution of the cavernomas appears to be the attraction of $Ccm3^{+/+}$ cells by $Ccm3^{-/-}$ cells, which, in turn, strongly contribute to the growth of the malformation.

An important question here is whether or not the occurrence of the malformations is a stochastic event, i.e. all endothelial cells have the potential to undergo clonal expansion and EndMT upon $Ccm3$ mutation in a random way. Alternatively, once mutated in $Ccm3$, only a subset of the cells might trigger the formation of lesions. This second possibility has been described in cancer, where tumour-initiating cells are responsible for metastatic dissemination and clinical relapse in a variety of cancers[43,50]. Tumour-initiating cells have similarities to normal tissue stem cells and undergo epithelial-to-mesenchymal transition. It is therefore attractive to hypothesise that, also in CCM, the

cavernomas are triggered by a few mutated endothelial cells with characteristics and markers specific of endothelial-resident progenitors, such as $Cd157^+$, $Procr^+$ and $PW1^+$. As in tumours, these cells would then recruit wild-type cells, which will, in turn, contribute to the growth of the cavernomas. Therefore, with these cavernoma-initiating cells normally present along the vessels, we propose a model in which upon loss of $Ccm3$ they undergo clonal expansion to form malformations. In a further step, the mutated cells recruit wild-type cells, which, in turn, strongly contribute to cavernoma growth (Fig. 9). Taking advantage of the $Procr^{CreERT2-IRES-tdTomato/+}$/$Ccm3^{f/f}$ mouse model, we have also shown that resident endothelial progenitors can trigger the formation of cavernomas upon deletion of $Ccm3$. We cannot completely exclude that these cavernoma-initiating cells are recruited systemically, as a bone marrow transplantation is not possible on new-born mice. However, the data presented here support the idea that cavernomas originate from $Procr^+$ progenitors, which have been described to reside in pre-existing vessels rather than being recruited[45] from circulating cells.

Broadly, our findings are suggestive of new possibilities for therapeutic intervention for patients with CCM, such as through inhibition of endothelial chemokines, to prevent the recruitment of wild-type endothelial cells or, more drastically, through identification and suppression of these cavernoma-initiating cells.

## Methods

**Antibodies and reagents**. Tamoxifen was from Sigma-Aldrich, and Vectashield without and with DAPI were from Vector Laboratories. The following primary

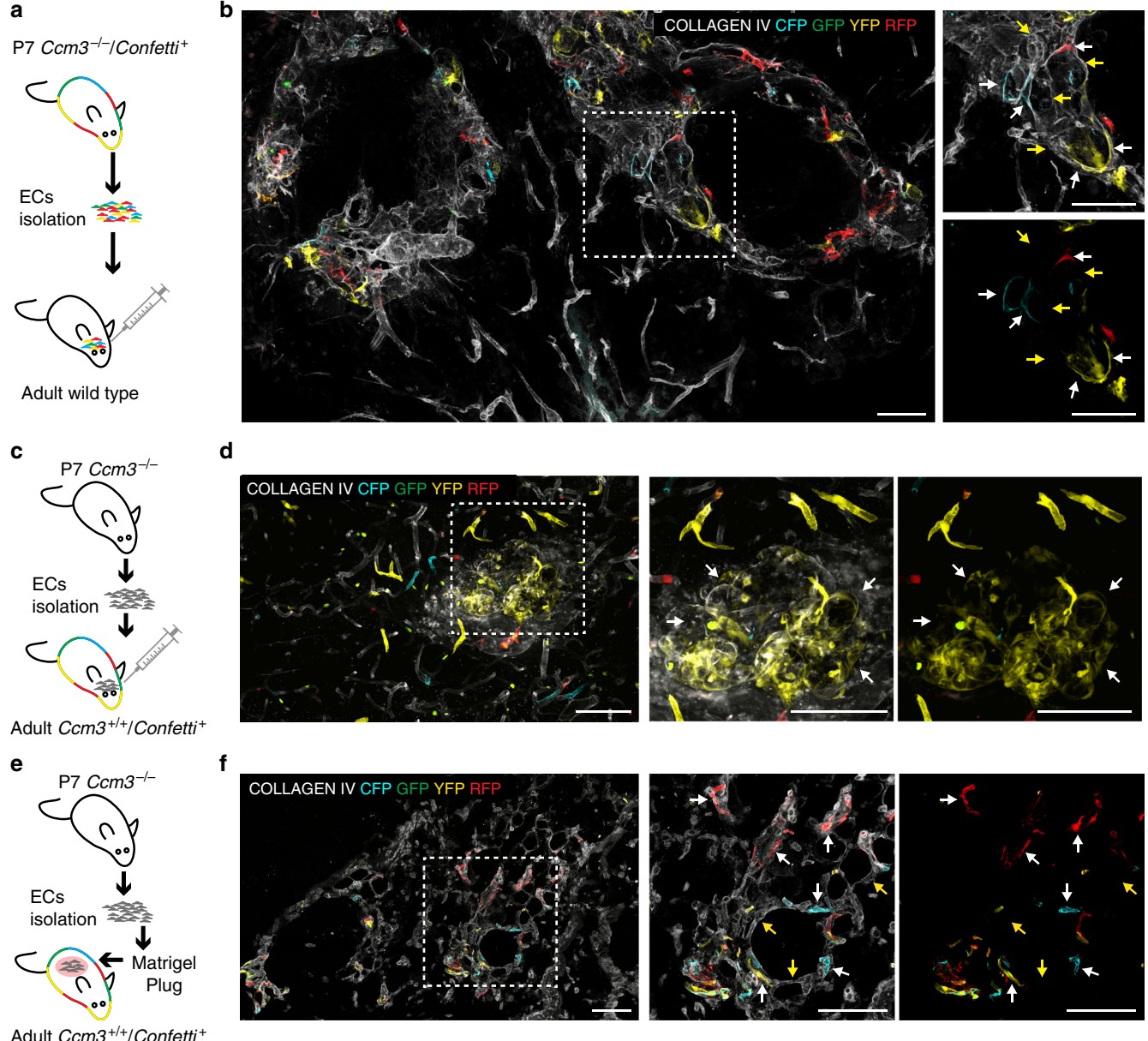

**Fig. 6** *Ccm3*$^{-/-}$ cells form cavernomas and recruit *Ccm3*$^{+/+}$ cells in wild-type animals. **a** Endothelial cells were isolated from acute *Cdh5(PAC)*-Cre-ER$^{T2}$/*Ccm3*$^{fl/fl}$/R26R-*Confetti* at P7 and re-injected into the brains of adult wild-type mice. **b** Brain section showing the formation of aberrant vessels that comprise both *Confetti*-positive endothelial cells, which are *Ccm3*$^{-/-}$ coming from the donor, as well as non-labelled *Ccm3*$^{+/+}$ endothelial cells coming from the host. **c** In the same way, endothelial cells were isolated from acute *Cdh5(PAC)*-Cre-ER$^{T2}$/*Ccm3*$^{fl/fl}$ at P7 and re-injected into the brains of adult *Cdh5(PAC)*-Cre-ER$^{T2}$/R26R-Confetti mice treated with 2 mg of Tamoxifen 1 week before injection. **d** Brain section showing the formation of aberrant vessels, which comprise also *Ccm3*$^{+/+}$ *Confetti*-positive endothelial cells coming from the host. **e** Endothelial cells were isolated from acute *Cdh5(PAC)*-Cre-ER$^{T2}$/*Ccm3*$^{fl/fl}$ at P7 and engrafted in a subcutaneous Matrigel plug into adult *Cdh5(PAC)*-Cre-ER$^{T2}$/R26R-*Confetti* mice. **f** Section of the plug showing the formation of aberrant vessels, which comprise both non-labelled *Ccm3*$^{-/-}$ endothelial cells coming from the donor as well as *Confetti*-positive endothelial cells, thus *Ccm3*$^{+/+}$ coming from the host. White arrows, endothelial cells positive for *Confetti*; yellow arrows, endothelial cells negative for *Confetti*. Scale bars, 100 μm

antibodies were used: Cd31 APC-conjugated anti-PECAM1 (1:50; 551262; BD Pharmingen); rabbit anti–phospho-Smad1/5 (Ser463/465) monoclonal (1:1000; 9516; Cell Signaling); rabbit anti-SMAD1 polyclonal (1:1000; 9743; Cell Signaling); rabbit anti–phospho-SMAD3 (Ser423/425) monoclonal (1:1000 for WB, 1:200 for IF; ab52903; AbCam); rabbit anti-SMAD3 monoclonal (1:1000; 9523; Cell Signaling); mouse anti-BMP6 (1:1000; Ab15640; AbCam); mouse anti-vinculin monoclonal (1:5000; V9264; Sigma-Aldrich); rabbit anti-Fsp1 polyclonal (1:200; 07-2274; Merck Millipore); Armenian hamster anti-Cd31 monoclonal (1:400; MAB1398Z; Merck Millipore); rat anti-Cd31 monoclonal (1:200; 553370 BD Biosciences); rabbit anti-Sca1 polyclonal (1:200; ab23750; AbCam); rat anti-Sca1 monoclonal (1:200; ab51317; AbCam); rabbit anti-Id1 polyclonal (1:200; sc-488; Santa Cruz); rabbit anti-Erg1/2/3 polyclonal (1:200; sc-353; Santa Cruz

Biotechnology); goat anti-Podocalyxin polyclonal (1:300; AF1556; R&D Systems); goat anti-Klf4 polyclonal (1:200; AF3158; R&D Systems); rabbit anti-collagen IV polyclonal (1:200; 2150-1470; Bio-Rad); rabbit anti-GFP polyclonal (1:300; A-6455; ThermoFisher); and mouse anti-α-SMA Cy3-conjugated monoclonal (1:500; C6198; Sigma-Aldrich); Goat anti-VE-Cadherin (C-19) (1:200; sc-6458; Santa Cruz Biotechnology). For immunofluorescence, the secondary antibodies were produced in either donkey or goat and targeted against the appropriate species, with conjugation with AlexaFluor 405, 488 or 647 or with Cy3 (1:400; Jackson Laboratories). For the whole-mount retina, the vessels were stained with biotiny-lated Isolectin B4 (1:100; B-1205; Vector Laboratories, Burlingame, CA, USA) and then with streptavidin conjugated with AlexaFluor 488, 555 or 647 (1:500; Invi-trogen). For western blotting, horseradish peroxidase (HRP)-linked anti-mouse

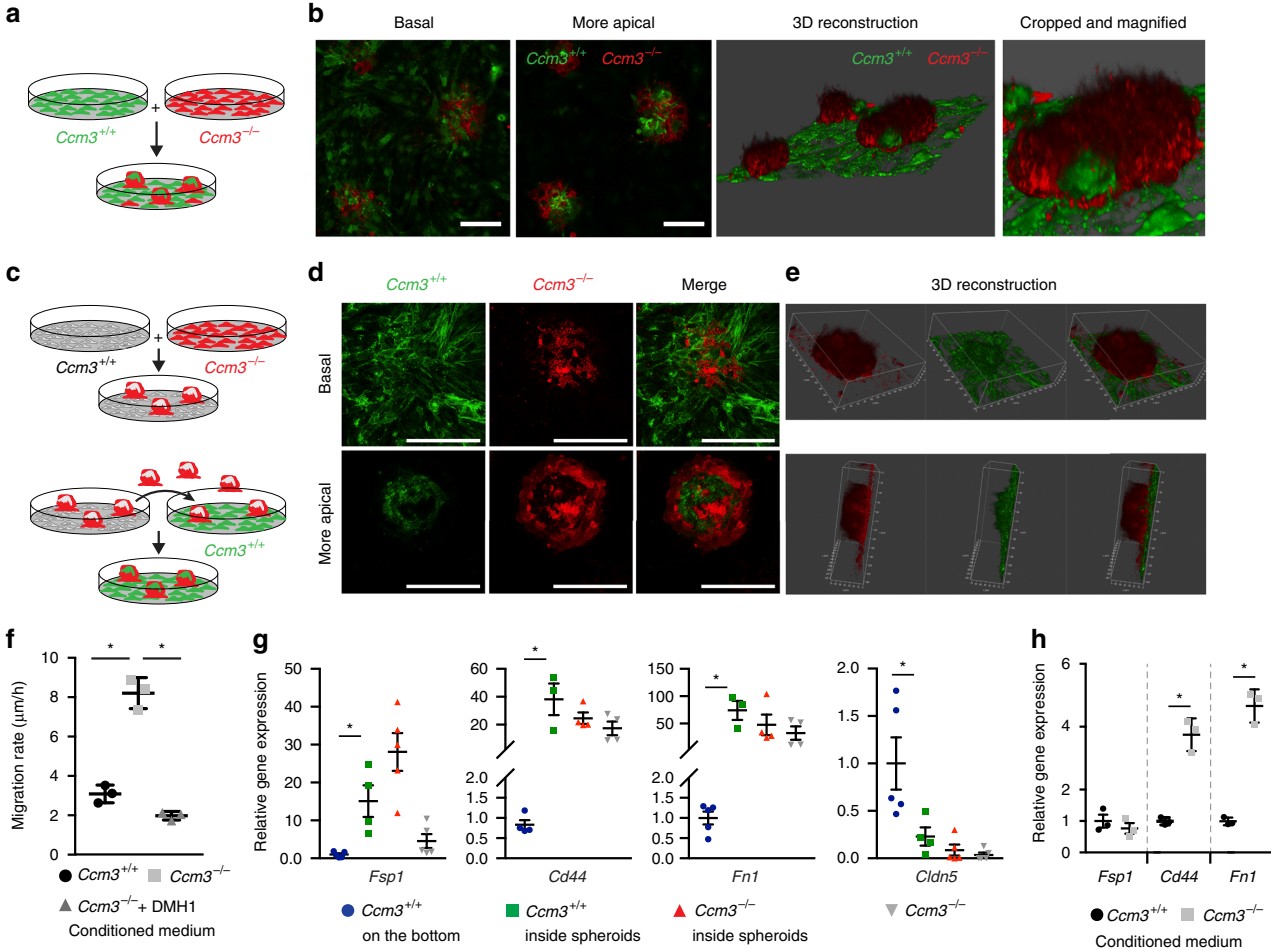

**Fig. 7** *Ccm3−/−* cells recruit and induce endothelial-to-mesenchymal transition in *Ccm3+/+* cells. **a, b** Lung immortalised *Ccm3−/−* endothelial cells expressing LifeAct-mCherry and *Ccm3+/+* cells expressing LifeAct-EGFP were co-cultured for 7 days, with spheroids formed, showing experimental scheme (**a**) and representative confocal images (**b**). **c** Non-labelled *Ccm3+/+* and LifeAct-mCherry expressing *Ccm3−/−* cells were co-cultured for 7 days, with the spheroids formed detached by shaking the culture plate. The detached spheroids were collected and added to an already formed monolayer of LifeAct-EGFP *Ccm3+/+* cells (see also Supplementary Fig. 7 for more details). **d** Representative confocal images of the basal and more apical levels of spheroids where *Ccm3−/−* (red) have recruited *Ccm3+/+* cells (green). **e** Representative three-dimensional reconstructions of spheroids with *Ccm3+/+* cells (green) recruited. Scale bars 50 μm (**b, d**). **f** Quantification of migratory rates of *Ccm3+/+* endothelial cells assessed using the classical wound-healing assay. After the scratch, cells were incubated with medium conditioned by either *Ccm3+/+* cells (negative control) or *Ccm3−/−* cells and without or with the BMP inhibitor DMH1. Data are means ± SE. $p < 0.001$ among groups (one-way analysis of variance (ANOVA)); *$p < 0.01$ (Tukey's post hoc test); $n = 3$ independent experiments. **g** Labelled *Ccm3+/+* and *Ccm3−/−* cells co-cultured as described above. After 7 days, spheroids were carefully detached, disaggregated and the cells separated by fluorescence-activated cell sorting (FACS). The cells that remained at the bottom were trypsinised and *Ccm3+/+* cells were separated by FACS. In parallel, *Ccm3−/−* cells were cultured separately as positive control and processed together with spheroids. After FACS, gene expression was analysed by RT-qPCR. Data are means ± SE; $p < 0.01$ among groups (one-way ANOVA); *$p < 0.01$ (Tukey's post hoc test); $n = 4$ independent experiments. **h** *Ccm3+/+* endothelial cells were incubated for 7 days with medium conditioned by either *Ccm3−/−* cells or *Ccm3+/+* cells (negative control), and the gene expression was analysed by RT-qPCR. Data are means ± SE. *$p < 0.01$ (Student's *t* test); $n = 3$ independent experiments. Source data are provided as a Source Data file

and anti-rabbit antibodies were used (1:2000; Cell Signaling). For the FACS analysis, the following primary antibodies were used: rat anti-Ki67 (1:100; 14-5698-82 clone SolA15, Invitrogen); purified mouse anti-BrdU (1:5; 347580, BD Bioscience); rabbit anti-PPH3 Ser10 (1:100; 06-570, Millipore).

The following reagents were also used: DMH1 (D8946; Sigma-Aldrich); K02288 (SML1307; Sigma-Aldrich), and recombinant BMP6 (507-BP-020; R&D systems); Propidium Iodide (PI; P4864, Sigma-Aldrich) and BrdU (B92285-1G, Sigma-Aldrich); Ribonuclease A from bovine pancreas (RNase A. cod.9001-99-4, Sigma-Aldrich), Erythrosin B, 0.1% solution in PBS (200964-25G, Sigma-Aldrich); Avertin (T48402; Sigma), Acrylamide (1610140; Biorad), VA-044, 2,2'-Azobis[2-(2-imidazolin-2-yl)propane]dihydrochloride (A3012; TCI chemicals), Microwave reaction vials (351521; Biotage), Microwave reaction vials cap with septum (352298; Biotage), Hystodenz (D2158; Sigma).

pmCherry-N1 vector was purchased from Clontec (Cat #632523). LifeAct-EGFP was a kind gift from Professor Wedlich-Söldner (WWU-Münster, Münster, Germany). FUGW, pCMV-delta R8.2 and pMD2.G were purchased from (Addgene, Cambridge, MA 02139, USA). The *Age*I, *Bsrg*I, *Xba*I and

*Nhe*I restriction enzymes were from New England Biolabs. Phalloidin-TRITC (Tetramethyl rhodamine Isothyocyanate) and Phalloidin-AlexaFluor488 were from Thermo Fisher Scientific.

The Lenti ORF clone of mGFP-tagged Human programmed cell death 10 (PDCD10), transcript variant 3 was purchased from (OriGene Technologies Inc., Rockville, MD 20850 USA).

**Mouse strains**. The following mouse strains were used: *Cdh5(PAC)*-Cre-ER[T2]/ *Ccm3*[f/f] mice in which *Ccm3*[f/f] mice with exons 4–5 of the *Ccm3* gene flanked by loxP sites (Taconic Artemis GmbH) were bred with *Cdh5(PAC)*-Cre-ER[T2] mice to obtain endothelial-specific and tamoxifen-inducible loss of function of the *Ccm3* gene, as previously described[24]. The *Cdh5(PAC)*-Cre-ER[T2] mouse line was kindly provided by R.H. Adams (Department of Tissue Morphogenesis, Faculty of Medicine, Max Planck Institute for Molecular Biomedicine University of Münster, Münster, Germany). *Cdh5(PAC)*-Cre-ER[T2]/*Ccm3*[f/f]/R26-EYFP mice were generated by crossing *Cdh5(PAC)*-Cre-ER[T2]/*Ccm3*[f/f] mice with the Rosa26-Stop[fl]-EYFP

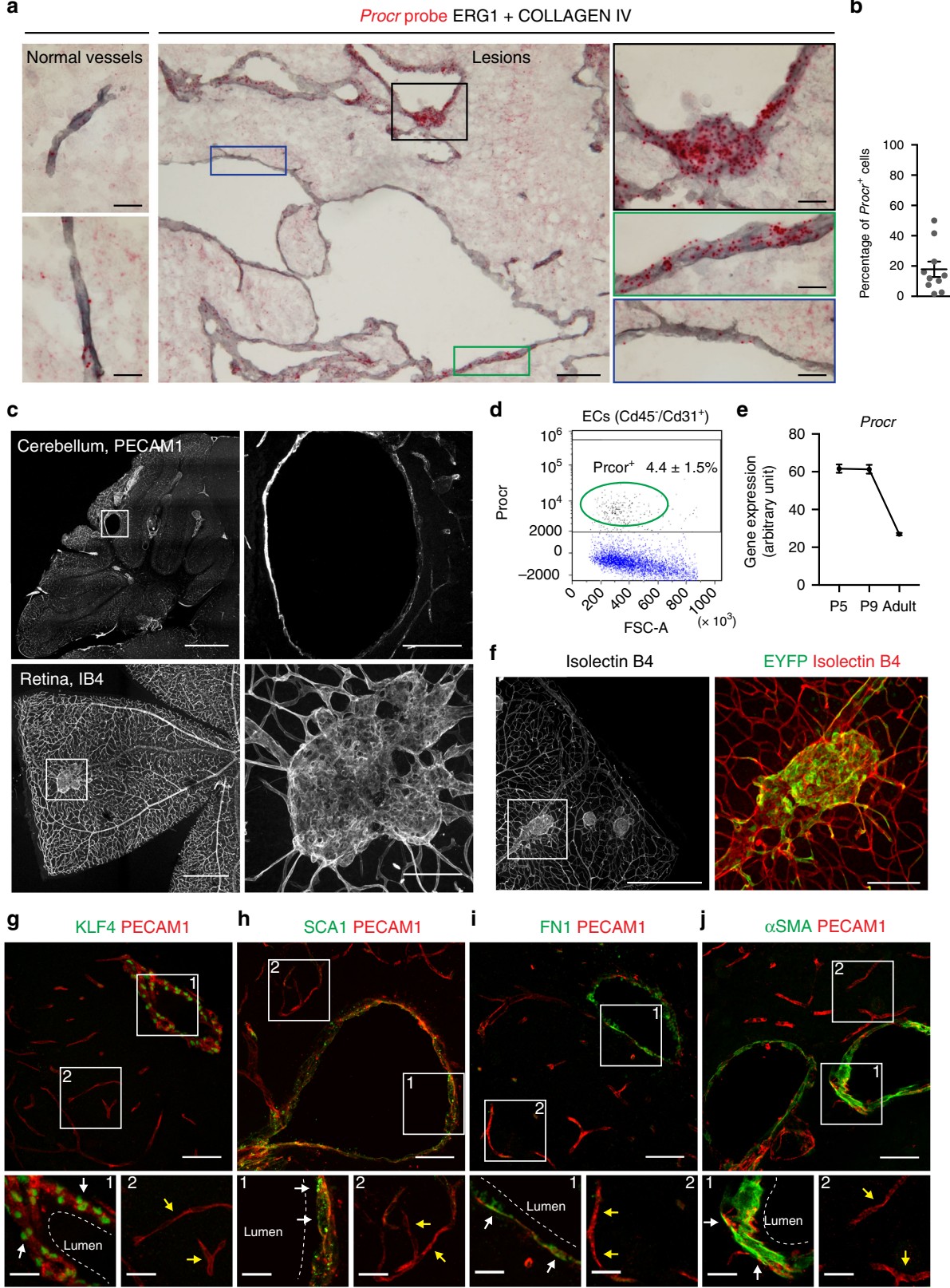

reporter mice[24,36–38], to monitor tamoxifen-induced Cre/loxP recombination. The Rosa26-Stop[fl]-EYFP mice were kindly donated by Dr. S. Casola (FIRC Institute of Molecular Oncology, Milan, Italy). For the *Cdh5(PAC)*-Cre-ER[T2]/*Ccm3*[f/f]/R26R-*Confetti* mice, *Cdh5(PAC)*-Cre-ER[T2]/*Ccm3*[f/f] mice were crossed with the stochastic multicolour reporter R26R-*Confetti* mice that allows clonal expansion by recombination of cells to be followed[21]. The Procr[CreERT2-IRES-tdTomato/+]/*Ccm3*[f/f] mice

were generated by crossing *Ccm3*[f/f] mice with Procr[CreERT2-IRES-tdTomato/+] knock-in mice, in which a CreERT2-IRES-tdTomato cassette is driven by the promoter of Procr[45,49], which allows induction of recombination in Procr-positive cells. Procr[CreERT2-IRES-tdTomato/+]/*Ccm3*[f/f] mice were then crossed with the Rosa26-Stop[fl]-EYFP reporter mice to generate Procr[CreERT2-IRES-tdTomato/+]/*Ccm3*[f/f]/R26-EYFP mice.

**Fig. 8** Endothelial progenitors are involved in cavernomas formation. **a** In situ hybridization for *Procr* using the RNAScope assay with chronic *Ccm3*$^{-/-}$ cerebral cavernous malformation (CCM) mouse brains. Red, *Procr* probe; grey, endothelial cells stained with a combination of anti-COLLAGEN IV and anti-ERG1 antibodies. Black and green panels show endothelial cells with high expression of Procr, blue panel shows cells with no expression of Procr. **b** Quantification of the percentage of endothelial cells that express *Procr* within the lesions; $n = 10$ lesions from 3 animals. **c** Procr$^{CreERT2-IRES-tdTomato/}$ $^{+}$/*Ccm3*$^{f/f}$ mice received tamoxifen injection at P1 and were analysed at P30. Representative images of the cerebellum stained for PECAM1 and whole-mount retina stained for Isolectin B4. **d** Fluorescence-activated cell sorting analysis on endothelial cells isolated from wild-type mice at P2. The plot shows the percentage of Procr$^{+}$ cells among the Cd45$^{-}$/Cd31$^{+}$ endothelial cells. Data are means ± SE; $n = 3$ animals. **e** RNAseq analysis on endothelial cells isolated from the brain of wild-type mice at different ages, showing the reduction of *Procr* expression during development. Data are means ± SE from 3 animals each group. Source data are provided as a Source Data file. **f** Procr$^{CreERT2-IRES-tdTomato}$/$^{+}$/*Ccm3*$^{f/f}$/R26-EYFP mice received tamoxifen injection at P1 and were analysed at P30. Representative images of whole-mount retina stained for Isolectin B4 and EYFP showing endothelial cells lining the lesion that underwent recombination. **g–j** Representative confocal images of brain sections double stained for PECAM1 (red) and the mesenchymal markers (green) KLF4 (**g**), SCA1 (**h**), FN1 (**i**) and αSMA (**j**). White arrows, cells lining the cavernoma; yellow arrows, cells in normal vessels. Scale bars, 500 μm (**c** and **f** main panels); 100 μm (**a** main panel, **c** and **f** magnifications), 50 μm (**g–j** main panels); 20 μm (**a**, **g–j** magnifications)

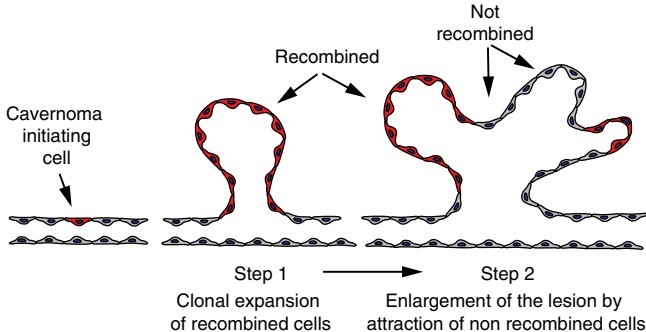

**Fig. 9** Model for the onset and growth of cavernomas. Cavernoma-initiating cells undergo clonal expansion in response to *Ccm3* loss and start to form cavernomas. In the further step, recombined cells recruit normal cells, which in turn, contribute to cavernoma expansion

**Mouse treatments**. All the procedures with the mice were performed in agreement with the Institutional Animal Care and Use Committee (IACUC) of FIRC Institute of Molecular Oncology, in compliance with the guidelines established in the Principles of Laboratory Animal Care (Directive 86/609/EEC) and approved by the Italian Ministry of Health.

Tamoxifen was dissolved in ethanol to 100 mg/ml and then diluted in corn oil to the final concentration of either 2 or 0.1 mg/ml. For the 'acute' model, 1-day-old mouse pups received a single intragastric injection of 50 μl 2 mg/ml tamoxifen (final dose: 100 μg per mouse). For the 'chronic' model, 2-day-old mouse pups received a single intragastric injection of 50 μl 0.1 mg/ml tamoxifen (final dose: 5 μg per mouse).

**Immunohistochemistry and confocal microscopy**. For the retinal vasculature, mouse eyes were fixed in 4% paraformaldehyde (PFA) for 2 h prior to dissection. Whole-mount retinas were incubated overnight with primary antibodies diluted in 10% donkey serum, 1% bovine serum albumin (BSA) and 0.5% Triton-X100 in phosphate-buffered saline (PBS) at 4 °C. After washes with 0.01% Triton-X100 in PBS, the retinas were incubated with the fluorophore-conjugated antibodies (Invitrogen) for 4 h at room temperature, washed, post-fixed with 4% PFA and mounted with Vectashield.

Immediately after dissection, mouse brains were fixed in 4% PFA overnight and then included in low-temperature melting agarose and 100-μm-thick sections were cut (VT1200s vibratome; Leica, Wetzlar, Germany). Free-floating sections were incubating overnight with the primary antibodies diluted in 5% donkey serum, 2% BSA and 0.3% Triton-X100 in PBS at 4 °C. After washes with 0.01% Triton-X100 in PBS, the sections were incubated with the fluorophore-conjugated antibodies (Invitrogen) for 4 h at room temperature, washed, post-fixed with 4% PFA and mounted with Vectashield.

Confocal microscopy was performed using a confocal microscope (TCS SP5, Leica), with the ImageJ software (NIH, New York, NY) used for image analysis.

**Lesion quantification**. For lesion quantification, 100-μm thick vibratome sections of brain were prepared and stained for endothelial markers as described above. For each brain, at least 10 sections were cut, and z-stacks were acquired with a ×10 objective. After maximal projection, the numbers and the areas of the lesions were manually measured by a blinded operator, using the ImageJ software.

**Passive tissue clearing**. Mice of strain and age as indicated in the text were anaesthetized by intraperitoneal injection of Avertin (20 mg/kg) and perfused with 4% PFA in PBS. The brains were carefully dissected out from the skull and post-fixed overnight by immersion in 4% PFA at 4 °C. The next day brains were washed in PBS and processed for sectioning.

Passive tissue clearing was carried out as previously described[51], with some modifications. Briefly, vibratome coronal sections of cerebellum (1-mm thick) were incubated overnight at 4 °C in hydrogel solution (4% Acrylamide, 0.25% VA-044 in PBS). Sections were then transferred in cold microwave reaction vials and hydrogel solution was replaced with fresh solution. Tubes were sealed using a cap with rubber septum. Before starting the polymerization, hydrogel solution was degassed by bubbling nitrogen through the septum for 2 min on ice. Samples were then placed at 37 °C to activate polymerization. After 2 h, excess acrylamide was removed and sections were incubated overnight in clearing solution (8% sodium dodecylsulfate (SDS), 0.01% Sodium Azide in water, pH 8.5) at 37 °C. The clearing solution was then replaced and passive clearing was continued for 72 h at 37 °C. The cleared sections were then washed in PBS containing 0.1% Triton X100 and 0.01% Sodium Azide (washing buffer) for 48 h, with at least four changes of washing buffer. To enhance both clearing and washing efficacy, the buffer-to-section volume ratio was always >50.

Immunostaining was performed at room temperature as follows. Sections were incubated with the indicated antibody in washing buffer supplemented with 5% donkey serum for 72 h. Samples were then washed for 30 h and incubated, for 72 h, with the appropriate secondary antibody (Alexa 647 conjugated) diluted in washing buffer. Finally, samples were washed for 30 h.

Sections were then incubated and, subsequently, mounted in refractive index matching solution (88% w/v Hystodenz in water) for confocal imaging with a Leica SP8 microscope. Three-dimensional reconstructions were created using the Arivis Vision4D software and movies were assembled with Photoshop.

**Brain endothelial cell isolation and FACS sorting**. Endothelial cells were isolated from the brains using a dissociator (gentleMACS; 130-093-235; Miltenyi) and the Adult Brain Dissociation Kits (130-107-677; Miltenyi), following the manufacturer's protocol. After dissociation, the cells were re-suspended in PBS with 2% foetal bovine serum and 2 mM EDTA and incubated for 30 min with an APC-conjugated anti-Cd31 antibody for FACS sorting. The sorting was performed on a MoFlo (Beckman Coulter Inc., Brea, CA, USA) cell sorter.

**Intracranial injection of cells and Matrigel plug assay**. Endothelial cells from either *Cdh5(PAC)*-Cre-ER$^{T2}$/*Ccm3*$^{f/f}$ or *Cdh5(PAC)*-Cre-ER$^{T2}$/*Ccm3*$^{f/f}$/R26R-Confetti mice at P7 were isolated as described above. After dissociation, endothelial cells were enriched by depletion of CD45-positive cells with CD45 MicroBeads (30-052-301; Miltenyi Biotech), followed by positive selection using CD31 MicroBeads (30-097-418; Miltenyi Biotech). The final cell pellets were washed with twice and then re-suspended in PBS.

For intracranial injection, 30,000cells/μl were re-suspended and 2 μl were injected into the brain of adult wild-type or *Cdh5(PAC)*-Cre-ER$^{T2}$/R26R-Confetti mice with a stereotaxic apparatus with the following coordinates: AP 1 mm, L 1 mm, D 2 mm. In parallel experiments, 500,000 cells isolated from P7 *Cdh5(PAC)*-Cre-ER$^{T2}$/*Ccm3*$^{f/f}$ mice were re-suspended in 500 μl of Matrigel and implanted subcutaneously to *Cdh5(PAC)*-Cre-ER$^{T2}$/R26R-Confetti adult mice.

**Indirect measure of cell proliferation: cell counting**. The endothelial cells *Ccm3*$^{+/+}$, *Ccm3*$^{-/-}$ and *Ccm3*$^{-/-}$ to which has been re-added the human *Ccm3* gene were plated at 60,000/cm$^2$ and counted at different time points (24, 48, 72 h) to assess their proliferation. After having been harvested, the cells were stained with the Erythrosin B and counted at an optical microscope through the Burker's chamber.

**FACS analysis: Ki67 and PI staining**. The protocol is intended for $3 \times 10^6$ cells and adapted from Vignon et al.[52]. After having been harvested, the cells were washed in PBS 1× and spun down at $500 \times g$ for 5 min. The pellet recovered was resuspended in PBS 1× and fixed by adding pure cold ethanol 100% (1 PBS 1×:3 EtOH 100%), dropwise while vortexing. The cells were left in fixative at least for 30 min on ice. Following two washes with PBS 1×, 1% FBS SA and 0.25% Triton X-100 (PFT), the cells were stained in 300 μl of anti-mouse/rat Ki67 (Invitrogen by Thermo Fisher Scientific, clone SolA15), diluted 1:100 in PFT, for 5 h at room temperature under constant rotation. A control tube was prepared with 300 μl of anti-rat 647 diluted 1:400 in PFT. After two washes with PFT, the cells were stained in 300 μl of PFT with anti-rat 647 antibody from Jackson, diluted 1:400, followed by incubation for 1 h at room temperature, in the dark, under constant rotation. Cells were washed twice with PBS 1×, centrifuged for 5 min at $500 \times g$ and resuspended in 1 ml of the intercalating fluorescent dye PI (2.5 μg/ml) plus RNase A (250 μg/ml). After an overnight incubation at 4 °C, FACS analysis was executed on the Attune NxT Life Technologies machine.

**FACS analysis: BrdU and PI staining**. The protocol is intended for $3 \times 10^6$ cells. While growing, the cells have been pulsed in MCDB131 medium containing 33 μM of BrdU. After having been harvested, the cells were washed in PBS 1× and spinned down at $500 \times g$ for 5 min. The pellet recovered was resuspended in PBS 1× and fixed by adding pure cold ethanol 100% (1 PBS 1× : 3 EtOH 100%), dropwise while vortexing. The cells were left in fixative at least for 30 min on ice. Following one wash with PBS 1×/BSA 1%, the cells were resuspended in 1 ml of denaturing solution (2 N HCl) and incubated at room temperature for 25 min. After addition of 3 ml of 0.1 M Sodium Borate ($Na_2B_4O_7$) and incubation for 2 min at room temperature, the cells were spinned down at $500 \times g$ for 5 min and washed twice with PBS 1×/ BSA 1%. The pellet was resuspended in 100 μl of anti-BrdU (purified mouse anti-BrdU, BD Biosciences, cod.347580), diluted 1:5 in PBS 1×/BSA 1%, for 2 h at room temperature, light protected, under constant rotation. A control tube was prepared with 300 μl of anti-mouse 488 diluted 1:400 in PBS 1×. After one wash with PBS 1×/BSA 1%, the cells were stained in 300 μl of PBS 1× with anti-mouse 488 antibody from Jackson, diluted 1:400, followed by incubation for 1 h at room temperature, in the dark, under constant rotation. Cells were then washed once with PBS 1×/BSA 1%, centrifuged for 5 min at $500 \times g$ and resuspended in 1 ml of the intercalating fluorescent dye PI (2.5 μg/ml) plus RNase A (250 μg/ml). After an overnight incubation at 4 °C, FACS analysis was executed on the Attune NxT Life Technologies machine.

**Endothelial cell culture and treatments**. For the in vitro experiments, the $Ccm3^{+/+}$ and $Ccm3^{-/-}$ immortalised lung endothelial cell lines were used[24]. The cells were grown in MCDB-131 (GIBCO, Invitrogen, Carlsbad, CA, USA) supplemented with 2 mM glutamine, 10% North America foetal bovine serum (HyClone), 20 μg/ml endothelial cell growth supplement (E2759, Sigma-Aldrich), heparin (40 μg/ml, from porcine intestinal mucosa; Sigma) and penicillin/streptomycin (100 units/l; Sigma). The starvation medium was MCDB-131 (GIBCO) with 2 mM glutamine and 1% BSA.

For experiments with BMP6 and inhibitors, the cells were starved overnight, pre-incubated with either 1 μM K02288[53] or 1 μM DMH1[36,39] for 1 h before stimulation with 100 ng/ml BMP6. The BMP6 and inhibitors were replaced daily over 5 days, and then the cells were lysed and analysed. In the same way, for the migration experiments, the cells were starved, pre-incubated with the inhibitors and then stimulated with BMP6. Just before stimulation, a wound was made, and after 48 h the cells were fixed and analysed.

For the experiments with conditioned medium, the cells were starved overnight and then incubated with complete medium that had been conditioned for 48 h by either the $Ccm3^{+/+}$ or $Ccm3^{-/-}$ cell cultures. The medium was replaced after 3 days, and the cells analysed after a further 3 days.

For the cell migration experiments, the cells were starved, pre-incubated with 1 μM DMH1 in medium for 1 h and then incubated with the conditioned medium from $Ccm3^{-/-}$ or $Ccm3^{+/+}$ cell cultures either without or with 1 μM DMH1. Just before stimulation, a wound was made, and after 48 h the cells were fixed and analysed.

**Cloning, recombinant lentiviruses and gene transduction**. The plasmid encoding LifeAct-EGFP was a kind gift from Professor Wedlich-Söldner[54]. To construct LifeAct-mCherry, the EGFP tag was excised by double digestion with the AgeI and BsrgI restriction enzymes, and the mCherry was ligated using the same sites. LifeAct-EGFP and LifeAct-mCherry were subsequently sub-cloned into the lentiviral vector (pFUGW). The plasmids containing the LifeAct-mCherry or LifeAct-EGFP fusion fragments were double-digested with XbaI and NheI, and subsequently the LifeAct-EGFP or LifeAct-mCherry fusions were gel-purified and sub-cloned into the lentiviral vector pFUGW using XbaI cloning sites. Recombinant lentiviruses were produced by triple transfection of the packaging 293T cells with the lenti FUGW carrying the gene of interest, packaging vectors (pCMV delta R8.2) and the envelope vector (pMD2.G) carrying the VSV glycoprotein (purchased from Addgene, Cambridge, MA 02139, USA). The virus was concentrated by ultracentrifugation at $50,000 \times g$ for 90 min at 4 °C, and the pellet was stored at −80 °C until use.

**Generation of cell lines**. To generate $Ccm3^{+/+}$ and $Ccm3^{-/-}$ cells expressing either LifeAct-EGFP or LifeAct-mCherry, $Ccm3^{+/+}$ and $Ccm3^{-/-}$ were transduced with the recombinant lentiviruses. In brief, the recombinant lentiviruses were re-suspended in serum-free MCDB-131 medium and added to the cells for 1 h at 37 °C.

To generate $Ccm3^{-/-}$ cells re-expressing mGFP-tagged CCM3, $Ccm3^{-/-}$ were transduced with the recombinant lentivirus Lenti ORF clone mGFP-tagged PDCD10 (OriGene Technologies Inc). In brief, the recombinant lentiviruses were re-suspended in serum-free MCDB-131 medium and added to the cells for 1 h at 37 °C. To increase the number of the cells, the cells were then passaged four times.

**$Ccm3^{+/+}$ and $Ccm3^{-/-}$ cell co-cultures**. $Ccm3^{+/+}$ and $Ccm3^{-/-}$ cells that expressed either LifeAct-EGFP or LifeAct-mCherry were mixed as single-cell suspensions (1:1) and plated and cultured under standard conditions. Usually the spheroids were analysed after 7 days. For cell isolation, the spheroids were removed from the monolayer following gentle shaking of the plate and collected and trypsinised. The single-cell suspensions obtained were immediately sorted by FACS for EGFP and mCherry expression, and then the cells were lysed and analysed for gene expression. For the experiments of spheroids detachment and transfer onto established monolayers, first the labelled $Ccm3^{-/-}$ cells were co-cultured with non-labelled $Ccm3^{+/+}$ cells. After 7 days, the established spheroids were detached by shaking and placed on the top of an established monolayer of labelled $Ccm3^{+/+}$ cells.

**Generation of endothelial spheroids**. Endothelial cell spheroids were generated as previously described[36].

Briefly, 3000 lung-derived immortalised endothelial cells were suspended in culture medium containing 0.20% (w/v) methyl cellulose and seeded in non-adherent round-bottom 96-well plates (Greiner). Under these conditions, all suspended cells contribute to the formation of one single rounded endothelial cell spheroid per well. These standardised spheroids were cultured for 6 days, then were harvested from 96-well plate, washed with PBS and processed for total RNA extraction.

**Wound-healing assay with two cell types and live imaging**. $Ccm3^{+/+}$ and $Ccm3^{-/-}$ cells that expressed either LifeAct-EGFP or LifeAct-mCherry were seeded in each side of an ibidi silicon insert and grown till confluence. Time-lapse images were acquired either immediately after the insert removal or after cells were allowed to close the wound. Live-cell imaging was performed with Leica TCS SP8 confocal (Leica, Germany) under 37 °C and 5% $CO_2$. Images were acquired using either HC PL APO 1.3 oil ×40 or HCX PL APO ×63 1.4 oil objective lenses (Leica, Germany), respectively.

**Gene expression analysis**. Total RNA was isolated using the RNeasy Mini Kits (Qiagen Inc., Santa Clarita, CA, USA) and 500 ng total RNA was reverse transcribed with random hexamers (High-Capacity cDNA Archive Kits; Applied Biosystems), following the manufacturer's instructions. The cDNAs were amplified using the TaqMan gene expression assay (Applied Biosystems, Foster City, CA, USA) and an ABI/Prism 7900 HT thermocycler. The following TaqMan Assays have been used: Klf4 mm00516104_m1; Fsp1 mm00803371_m1; Cd44 mm01277163_m1; Fn1 mm01256734_m1; Cldn5 mm00727012_s1; Ccm3 mm00727342_s1; Sca1 mm00726565_s1; Id1 mm00775963_g1; and as housekeeping genes: 18s Hs99999901_s1; Gapdh mm99999915_g1; and Hprt1 mm00446968_m1.

**Western blotting**. The cells were lysed in boiling modified Laemmli sample buffer (2% SDS, 20% glycerol, 125 mM Tris-HCl, pH 6.8). Equal amounts of protein were loaded onto gels and separated by SDS–polyacrylamide gel electrophoresis and then transferred to nitrocellulose membranes (Protran; Whatman). After blocking with 5% BSA and incubations with the primary and HRP-linked secondary antibodies, specific binding was quantified using a chemiluminescence system (GE Healthcare).

**In situ hybridisation**. In situ hybridisation was carried out on brain cryosections (RNAScope assay; AcdBio), following the manufacturer's recommendations. The Mm-Procr probe (410321) was used for the hybridisation of Procr, followed by detection (RNAScope 2.5 HD Detection Reagents – RED; 322360).

**Statistical analysis**. Statistical significance was determined using a one-way analysis of variance (ANOVA) tests followed by Tukey's post hoc analysis. For the growth curve experiment, two-way ANOVA test were followed by Tukey's post hoc analysis for multiple comparison within each time point.

In Fig. 7h and Supplementary Figs. 6e and 9, the statistical significance was determined using a non-paired two-tailed $t$ test, assuming unequal variances.

**Reporting summary**. Further information on research design is available in the Nature Research Reporting Summary linked to this article.

## Data availability

All the data supporting the findings of this study are available from the corresponding authors upon reasonable request. The source data underlying Figs. 1c, 2f–h, 3e–g, 4b, 7f–h and 8c–e and Supplementary Figs. 1a, b, e, 2a, b, 3a, b, 5a, b, 6e, 9 and 10a–d are provided as a Source Data file.

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

## Acknowledgements

This study was supported by *Associazione Italiana per la Ricerca sul Cancro* (AIRC IG 18683) and AIRC 5×1000 call 'Metastatic disease: the key unmet need in oncology' to MYNERVA project, #21267 (MYeloid NEoplasms Research Venture Airc), by The Swedish Research Council, contract No. 2013-9279, and the Knut and Alice Wallenberg Foundation, by the European Research Council (project EC-ERC-VEPC, contract 742922), by Initial Training Networks (ITN) Brain Barrier Training (BtRAIN), contract No. 675619, by CARIPLO Foundation project : 2014-1038 and Project 2016-0461, Be Brave for Life Foundation Micro Grant. We are grateful to Dr. Maria Grazia Totaro and Dr. Arianna Quintè of the Cytometry Unit of Cogentech for excellent technical support. We thank Dr Christopher P. Berrie for editorial assistance. We are indebted to Professor Ralph Adams for providing Cdh5(PAC)-CreERT2 mice.

## Author contributions

M.M. and E.D. conceived the project and designed the experiments; M.M. and C.M. performed in vivo and in vitro experiments; A.A.T., M.C. and M.V. performed in vitro experiments; F.P. performed histological experiments; F.O. performed confocal imaging and assembled figures; Q.C.Y. and Y.Z. provided Procr$^{\text{CreERT2-IRES-tdTomato/+}}$ mice; M. G., M.G.L. and P.U.M. contributed to scientific discussion; C.F. and P.G. provided reagents; M.M. and E.D. wrote the manuscript, which has been revised by all the authors.

## Additional information

**Competing interests:** The authors declare no competing interests.

