## [Peer Review File · Nature Communications]

Reviewers' comments:

Reviewer #1 (Remarks to the Author):

Cerebral cavernous malformation is a severe neurovascular familial or sporadic disease that is characterised by capillary

venous cavernomas, and is due to loss of function mutations to one

of three CCM genes. The authors previously noted that in sporadic cavernomas only a small fraction of endothelial cells shows mutated CCM genes and that in a mouse model of the human disease, endothelial cells lining the lesions have different transcriptome features from much of the surrounding endothelium.

In this report, the authors state that cavernomas originate from clonal expansion of a small number of Ccm3-null endothelial cells

that express both endothelial and mesenchymal/stem cell markers. Second, this small population of mutant cells is purported to attract surrounding wild-type cells to induce them to express

mesenchymal/stem-cell markers. The authors conclude these characteristics of Ccm3-null cells are reminiscent of 'tumour initiating cells' that are responsible for tumour growth and metastatic differentiation and that CCM has benign tumour characteristics. These are the main claims.

There are several points that need to be addressed to solidify the claims:

1) How do the authors define a clone in the different tissues and at different ages of the mice? This is a key point. Is the definition based on a certain number of contiguously expressing cells within a vessel: how many cells comprise a clone (line 90-95; 123-130)?

2) While some evidence for the ability of the mutant cells to modify the behavior of the wild-type cells in vitro, it was not proven that the mutant cells possess the capacity to recruit and cause the formation of CCM in vivo. This could be readily accomplished using simple Matrigel plug assays containing the mutant cells (labeled) implanted in the Cdh5-reporter mice to distinguish the mutant from wild type cells. Generation of cavernous hemangiomas in the plugs would prove the mutant cells were indeed capable of recruiting and causing these lesions (without the recruited cells becoming genetically modified).

3) (lines 165-175) Do the recruited wild-type cells lose their endothelial functional capacities (vessel forming potential) when they upregulate the EMT profile of genes? If the cells that upregulate the EMT genes are simply left in culture with removal of the mutant genes, do the wild-type cells revert back to a normal endothelial phenotype and function?

Reviewer #3 (Remarks to the Author):

Malinverno et al describe clonal endothelial cells (ECs) that contribute to the pathobiology of Cerebral Cavernous Malformations (CCM). The study is a followup from several recent reports from the Dejana lab describing ECs with mesenchymal-like features in CCM. The authors use an elegant reporter system (R26R-Confetti mice) in mice with conditional CCM3 deletion in ECs to demonstrate clonal expansion of ECs in early lesions which eventually become “mixed” due to recruitment of ECs that are CCM3^{+/+}. The authors suggest that CCM3 null ECs recruit CCM wildtype ECs into CCM lesions via BMP6. They also suggest that CCM mutations within a small group of EC progenitors are primarily responsible for driving the formation of CCM lesions although this data needs further elaboration. Overall, the study presents a compelling model for a “field effect” in CCM whereby CCM3^{-/-} ECs can alter the behavior of CCM3^{+/+} ECs nearby thereby contributing to CCM progression.

Specific critique (in no particular order)

The authors should reconsider the use of the word “transform” as in “...CCM3^{-/-} ECs transform CCM3^{+/+} ECs...”. “Transform” generally refers to mutations that enable the acquisition of malignant traits and the authors do not demonstrate that the cells are indeed “transformed” in this way. A different term might be chosen to make this distinction.

With regards to clonal expansion of CCM3^{-/-} ECs: the evidence for this is based on color labeling in the confetti background; however, because labeling is stochastic, it is theoretically possible that two neighboring cells are labeled with the same color fluorophore but are not actually derived from the same parent cell. The authors should in the least discuss this possibility or provide additional experimental evidence that the ECs in CCM lesions are indeed clonal.

It is not clear why the particular genes assessed (Klf4, Id1, pSMAd3, Sca1, etc) are called EndMT markers - what distinguishes these genes as EndMT markers? Aren't these genes expressed by normal ECs at some level even basally? The immunofluorescence staining in fig 3e-h might be more convincing with normal vessels as a comparison and help indicate whether these markers typify EndMT or are enriched in areas where EndMT is or has occurred.

Rescue experiments should be carried out using CCM3 re-expression in CCM3^{-/-} ECs. Does this suppress EndMT marker genes or correct their (mal)functional properties?

What is the fate/function of CCM3^{-/-} ECs in vivo? The authors should re-inject these cells into syngeneic mice to assay their morphology and function (e.g. anastomosis/perfusion). Do they form aberrant vascular structures when re-injected? These experiments could be done using CCM3 rescue in CCM3^{-/-} ECs (see above).

Since the authors suggest that CCM3^{-/-} ECs expand clonally in the CCM lesions, is it assumed that this expansion is via proliferation? If so, what is the link between CCM3 and cell cycle? Is deletion of CCM3 sufficient to drive S-phase or inhibit apoptosis in ECs? The authors should test this possibility experimentally.

The endothelial progenitor marker staining (CD157/CD200) is difficult to see and needs quantification. This staining also looks perivascular? Related to this, what fraction of Procr⁺ cells are labelled? Staining for Procr should be carried out. The concept that a small number of EC progenitors are the “drivers” of CCM lesions is compelling, but the data needs to be presented more clearly.

Can the authors assay the CCM gene product (PDCD10) and show mosaicism in human CCM lesions (PDCD10⁺ ECs and PDCD10⁻ ECs)?

Do culture expanded CCM3^{-/-} ECs retain or show enrichment for EC progenitor markers compared to their CCM3^{+/+} counterparts?

The authors suggest that CCM3^{-/-} ECs recruit/convert CCM3^{+/+} ECs, but do CCM3^{+/+} ECs block/revert/normalize CCM3^{-/-} ECs in any way? This should be tested experimentally.

Is there any evidence that CCM3^{+/+} ECs are recruited systemically (i.e. bone marrow)? Or are these locally-derived ECs from nearby vessels? A bone marrow origin is doubtful but maybe a point worthy of discussion if the authors have an opinion or data.

Reviewer #4 (Remarks to the Author):

The work by Malinverno and colleagues builds upon previous findings by the Dejana laboratory suggesting that endothelial cells (ECs) lining cavernomas, the hallmark vascular lesions of the cerebrovascular disorder Cerebral Cavernous Malformations (CCM), are morphologically and molecularly different from ECs of the same vessel outside the lesion site. Familial forms of CCM are underlain by mutations in one of three genes (CCM1-3), and the two-hit mechanism (a germline mutation followed by a somatic mutation in a small number of ECs) has been employed to explain disease pathogenesis. In this manuscript, the authors test the hypothesis that CCM lesions originate from a small number of Ccm3-mutant ECs, which expand clonally, recruit surrounding wild type ECs, and induce their molecular and morphological transformation.

The manuscript addresses an interesting and novel idea, which is tested using acute and chronic mouse models of CCM3 disease, additional inducible transgenic mice, as well as in vitro co-culture assays of wild type or Ccm3 mutant ECs. The authors present a wealth of diverse and detailed data arguing in support of the proposed hypothesis. The study has considerable strengths, but also significant weaknesses. The major strength of the study is the novelty of the idea tested, namely, that clonal expansion of ECs mechanistically explains cavernoma formation. The major weakness, on the other hand, is that the conclusion that cerebral cavernomas arise from clonal expansion of ECs (as also stated in the title) is not fully supported by the data.

Below are the major points I wish to raise about the study:

1) Figure 1

The authors should include a picture of wild type retina, and not of (or, in addition to) wild type brain (current panel 1a). If the goal is to illustrate the clonal nature of retinal lesions, then the reference should be an image of the wild type retina. Is it possible that cavernomas have clonal origin because they arise from a vessel that is itself clonal?

The lesions (magnification panel 1) in panels 1c and 1c' are shown as being representative of a "larger" lesion; however, they do not have the appearance of mature lesions with multiple caverns. Moreover, a section through a (any) lesion is not necessarily perpendicular to the wall of the dilated vessel (as it could be at an angle). Consequently, without a true 3D reconstruction, (and not just assessment of 10 thin sections [unclear if they are serial]), lesion area and diameter cannot be correctly assessed. This is an important point for the premise of the study, as small diameter lesions are said to be clonal, whereas, in contrast, large cavernomas are said to be mosaic. This point becomes even more relevant as it is difficult to assess the non-clonality of the large lesions presented in panel 1d (too few cells of a different origin compared to the vast majority of the lesion that appears to be rather clonal).

2) Extended Figure 1a

Please correct the schematic (it is indicated lethal at P10, however the text -line 98- states “lethal in around 12 days”

3) Figure 2d, e and Extended Figure 1d (text line 104)

The comparison attempted here is not valid – the acute model (P8) cannot be compared to the chronic model at P30. The comparison should be carried out within the same model.

4) In order to conclude that cavernomas form via progression from a clonal, small/immature/early lesion to a non-clonal, larger/mature lesion (in which wild type cells are being recruited and induced to change in the presence of Ccm3-mutant cells), lesion progression (and accompanying molecular changes) should be documented in the chronic models. Specifically, staining with EMT markers over time will be necessary on smaller lesions that are 100% positive vs. larger (more mature) lesions, which should be mosaic. The authors performed a similar comparative analysis focusing on KLF4-positive cells (Fig. 2e), which should be extended to earlier time points in the same (chronic) model.

5) In Figure 2f-m (presumably corresponding to the slow progression/chronic model – please clarify in the legend), the quality of the immunofluorescence could be improved. PECAM1 staining appears not to outline the lining of the lesions in its entirety; as a consequence, some, presumably, ECs appear positive for the mesenchymal markers, but lack PECAM1 staining. Can the authors show in the same model that smaller/immature/clonal lesions are 100% positive for EMT markers, whereas larger/mature lesions show a significant reduction of ECs that are positive for EMT markers?

6) Extended Figure 2c

The in situ hybridization data are not convincing: this figure could be improved by DIC microscopy to visualize the outline of the cells; please comment on the staining outside the lesions.

7) Extended Figure 3

Please specify that these are immortalized lung endothelial cells

8) Figure 4

i) Panels b, d, and e: it is very difficult to assess the number of wild type ECs recruited in the Ccm3-mutant spheroids.

ii) Panel f and lines 174 and 189. The observed changes in migration rate in response to conditioned media from Ccm3-mutant EC cultures are mild and do not support the conclusion of the involvement of secreted factors (see also points iii and iv below).

iii) Panel f: There is a discrepancy between the data on the rate of migration described in this panel (3 $\mu\text{m/hr}$ with conditioned media from wild type cultures vs. 8 $\mu\text{m/hr}$ with conditioned media from Ccm3-mutant cultures) and data in Extended Figure 7e (7 $\mu\text{m/hr}$ with vehicle as negative control). How do the authors explain this?

iv) Line 162 and Extended Figure 6. As the authors mention, involvement of cell-cell contacts in mediating the recruitment of wild type cells by Ccm3-mutant ECs is indeed plausible.

v) Panel g (and lines 165-168): the comparison here is between wild type cells isolated from the spheroids vs. wild type cells cultured as a monolayer. This is problematic, and a more accurate comparison would have been with wild type cells that have not been recruited/encapsulated in the spheroids, yet cultured in the same dish. An additional problem is that by definition cells that grow in 3D (such as within a spheroid), undergo morphological (and molecular) changes that distinguish them from the same cells grown as a monolayer.

9) line 170: how do the authors explain the increase on Fsp1 relative expression only in the presence of BMP6, but not upon exposure to conditioned media from Ccm3-mutant cultures?

10) Although the source of the different materials used in every figure/panel is described in Methods, having to constantly refer to the Methods to retrieve this information is disruptive. It would be preferable if this information is also provided as needed throughout the text and in the figure legends (e.g. ECs in culture are either primary brains ECs or lung ECs; FACS is performed not on mice, but specifically on brains, etc.)

In summary, the work presented in this manuscript is intriguing and proposes a novel mechanism to explain the formation of CCM lesions. However, additional experiments are needed to support the conclusions reached by the authors, as well as the definitive nature of the title-statement, which in this reviewer's opinion is very strongly worded, without being fully substantiated by the findings as these are currently detailed, and thus should be revised.

We would like to thank the reviewers for the careful analysis of our paper and the precious comments and suggestions that helped us to improve our work.

We have carefully taken into consideration the points raised and performed the requested additional experiments adding more data to the original manuscript. We enclose below a point-by-point reply to the reviewers' criticisms.

Please note that in the present version the manuscript comprises 6 new figures and 8 movies.

In the text, changes have been highlighted with red font.

Reviewers' comments:

Reviewer #1 (Remarks to the Author):

Cerebral cavernous malformation is a severe neurovascular familial or sporadic disease that is characterised by capillary venous cavernomas, and is due to loss of function mutations to one of three CCM genes. The authors previously noted that in sporadic cavernomas only a small fraction of endothelial cells shows mutated CCM genes and that in a mouse model of the human disease, endothelial cells lining the lesions have different transcriptome features from much of the surrounding endothelium. In this report, the authors state that cavernomas originate from clonal expansion of a small number of Ccm3-null endothelial cells that express both endothelial and mesenchymal/stem cell markers. Second, this small population of mutant cells is purported to attract surrounding wild-type cells to induce them to express mesenchymal/stem-cell markers. The authors conclude these characteristics of Ccm3-null cells are reminiscent of "tumour initiating cells" that are responsible for tumour growth and metastatic differentiation and that CCM has benign tumour characteristics. These are the main claims.

There are several points that need to be addressed to solidify the claims:

1) How do the authors define a clone in the different tissues and at different ages of the mice? This is a key point. Is the definition based on a certain number of contiguously expressing cells within a vessel: how many cells comprise a clone (line 90-95; 123-130)?

In the original manuscript, we defined a lesion as "clonal" when composed of at least 80% of cells of the same color. The predominance of one color on the others suggests that a clonal expansion has occurred (Manavski Y. et al. Cric Res 2018). However, to better define a clone, as suggested by the reviewer, we performed a more quantitative analysis. We analysed the cerebellum from *Cdh5(PAC)-Cre-ER^{T2}/Ccm3^{fl}/R26R-Confetti* mice and their *Ccm3^{+/+}* counterpart after injection of low tamoxifen at P2. We consider a clone any group of contiguous cells expressing the same color within a vessel and counted the number of cells that compose the clone. In the vessels of *Ccm3* WT mice, the average size of clones was 1.14 ± 0.016 cells, with 88% of single cells and no clones larger than 4 cells. Likewise, in normal vessels of the *Cdh5(PAC)-Cre-ER^{T2}/Ccm3^{fl}/R26R-Confetti*, the average size of clones was 1.19 ± 0.014 cells, with 85% of single cells and no clones larger than 4 cells. Within the lesions instead, the clones showed an average size of 9.35 ± 0.867 cells, with 43% of clones larger than 4 cells. Moreover, around 12% of clones was formed by 10 up to 60 cells. A very small fraction of clones of more than 1 cell (i.e. two or three cells) within WT vessels could be expected, as recombination occurred at P2 while the vasculature is still developing and growing, and a rate of one or two cell divisions in the endothelium is physiological. The presence of clones of significantly larger size, i.e. 10 or more cells, can strongly support the concept that an unusual expansion has occurred. These new data have been inserted in the text at lines 141-160 page 6-7 and in the Fig. 3d-f and Extended Data Fig 4.

2) While some evidence for the ability of the mutant cells to modify the behavior of the wild-type cells in vitro, it was not proven that the mutant cells possess the capacity to recruit and cause the formation of CCM in vivo. This could be readily accomplished using simple Matrigel plug assays containing the mutant cells (labeled) implanted in the Cdh5-reporter mice to distinguish the mutant from wild type cells. Generation of cavernous hemangiomas in the plugs would prove the mutant cells were indeed capable of recruiting and causing these lesions (without the recruited cells becoming genetically modified).

We followed the suggestion of the reviewer and implanted a Matrigel plug with *Ccm3^{-/-}* ECs in *Ccm3^{+/+}* mice, in order to prove their capacity to recruit WT cells and generate cavernomas in vivo. In particular, we isolated ECs from brains of P7 *Cdh5(PAC)-Cre-ER^{T2}/Ccm3^{fl}* mice injected with full dose of tamoxifen and implanted (in a Matrigel plug) in *Cdh5(PAC)-Cre-ER^{T2}/Ccm3^{fl}/R26R-Confetti* mice. After 10 days, we could detect abnormal vessels composed also by Confetti positive ECs coming from the host. This supported the capacity of the *Ccm3* null cells to recruit WT cells and to form cavernomas *in vivo*.

With the aim of further prove the capacity of *Ccm3* null ECs to generate cavernomas in a WT context, we isolated ECs from *Cre-ER^{T2}/Ccm3^{fl}/R26R-Confetti* mice and re-injected into brains of WT adult mice. After only 4 days, Confetti positive formed cavernomas, which also comprised WT non-labelled cells. In parallel,

we performed the reverse experiment and isolated ECs from non-labelled *Ccm3*^{-/-} mice and injected into brains of *Ccm3*-WT but Confetti positive adult mice. Also in this case, KO cells generated abnormal vessels with the participation of Confetti-positive WT cells from the host. This further confirmed the capacity of *Ccm3* null ECs to generate cavernomas recruiting WT ECs. These new data have been inserted in the text at lines 200-213 page 9 and Fig. 6.

3) (lines 165-175) Do the recruited wild-type cells lose their endothelial functional capacities (vessel forming potential) when they upregulate the EMT profile of genes? If the cells that upregulate the EMT genes are simply left in culture with removal of the mutant genes, do the wild-type cells revert back to a normal endothelial phenotype and function?

This is an interesting point raised by the reviewer, whether the induction of EndMT in WT cells is, or not, a reversible event. In order to answer this question, and following the suggestion of the reviewer, we repeated the experiment and sorted WT cells from the spheroids and the underlying monolayer, and then left them in culture for 7 days separately. After 7 days in culture, WT ECs isolated from the spheroids reverted to a normal phenotype and down regulated the EndMT makers to a level comparable to wild type cells coming from the underlying monolayer. This suggests that the induction of EndMT requires constant action of *Ccm3*^{-/-} cells, and that once the WT cells are cultured alone can revert to normal phenotype. This new data have been added in the text at lines 244-246 page 10 and Extended Data Fig. 9.

Reviewer #3 (Remarks to the Author):

Malinverno et al describe clonal endothelial cells (ECs) that contribute to the pathobiology of Cerebral Cavernous Malformations (CCM). The study is a followup from several recent reports from the Dejana lab describing ECs with mesenchymal-like features in CCM. The authors use an elegant reporter system (R26R-Confetti mice) in mice with conditional CCM3 deletion in ECs to demonstrate clonal expansion of ECs in early lesions which eventually become "mixed" due to recruitment of ECs that are CCM3^{+/+}. The authors suggest that CCM3 null ECs recruit CCM wildtype ECs into CCM lesions via BMP6. They also suggest that CCM mutations within a small group of EC progenitors are primarily responsible for driving the formation of CCM lesions although this data needs further elaboration. Overall, the study presents a compelling model for a "field effect" in CCM whereby CCM3^{-/-} ECs can alter the behavior of CCM3^{+/+} ECs nearby thereby contributing to CCM progression.

Specific critique (in no particular order)

The authors should reconsider the use of the word "transform" as in "...CCM3^{-/-} ECs transform CCM3^{+/+} ECs...". "Transform" generally refers to mutations that enable the acquisition of malignant traits and the authors do not demonstrate that the cells are indeed "transformed" in this way. A different term might be chosen to make this distinction.

We agree with the reviewer that the term "transform" can refer to malignant behaviour of cancer cells, thus not properly used in the CCM context. Therefore, we substituted "transform" with "induce their phenotypical modifications", as in line 198 page 9.

With regards to clonal expansion of CCM3^{-/-} ECs: the evidence for this is based on color labeling in the confetti background; however, because labeling is stochastic, it is theoretically possible that two neighboring cells are labeled with the same color fluorophore but are not actually derived from the same parent cell. The authors should in the least discuss this possibility or provide additional experimental evidence that the ECs in CCM lesions are indeed clonal.

As pointed out by the reviewer, the stochastic nature of the recombination in the Confetti system, cannot exclude that by chance two adjacent cells get the same colour even if they do not derive from the same cell. To better assess the frequency of this event, we performed a more quantitative analysis on the low tamoxifen recombined model. We analysed cerebellum from Cre-ER^{T2}/*Ccm3*^{fl}/R26R-Confetti mice and their *Ccm3*^{+/+} counterpart after injection of low dose of tamoxifen at P2. We consider a clone any group of contiguous cells expressing the same color within a vessel and counted the number of cells that compose the clone. In the vessels of WT mice, the average size of clones was 1.14 ± 0.016 cells, with 88% of single cells and no clones larger than 4 cells. Likewise, in normal vessels of the VECPC/CCM3/Confetti mice, the average size of clones was 1.19 ± 0.014 cells, with 85% of single cells and no clones larger than 4 cells. Within the lesions instead, the clones showed an average size of 9.35 ± 0.867 cells, with 43% of clones larger than 4 cells. Moreover, around 12% of clones had a size from 10 up to 60 cells. A small fraction of clones of more than 1 cell (i.e. two or three cells) within WT vessels could be explained by either cell division (which can be expected in a developing vasculature) or the chance that two adjacent cells got the same colour.

The presence of clones of significantly larger size, i.e. 10 or more cells, however, strongly support the concept that an unusual expansion has occurred. Therefore, although unlikely, we cannot exclude that a clone originates from two cells of the same colour instead of one. However, even if a clone composed of 20 cells originated from two original cells of the same color, still, we believe that this condition requires in any case an unusual increase in clonal expansion.

These new data have been inserted in the text at lines 141-160 page 6-7 and in the Fig. 3d-f and Extended Data Fig 4.

It is not clear why the particular genes assessed (Klf4, Id1, pSMAd3, Sca1, etc) are called EndMT markers - what distinguishes these genes as EndMT markers? Aren't these genes expressed by normal ECs at some level even basally? The immunofluorescence staining in fig 3e-h might be more convincing with normal vessels as a comparison and help indicate whether these markers typify EndMT or are enriched in areas where EndMT is or has occurred.

Some of these markers are expressed at a low level in normal ECs, such as *Klf4*, *Sca1*, *Id1*, *Cd44* as revealed by RT-qPCR on freshly isolated ECs (see Maddaluno et al. Nature 2013 and Bravi et al. PNAS 2015). Other markers such as Alpha-Smooth Muscle Actin are not detectable in resting endothelial cells as published in Maddaluno et al and in Bravi et al. but are upregulated in CCM deficient cells. When cells undergo EndMT they up-regulate the expression of these genes if compared to WT ECs and partially down regulate endothelial specific markers such as VE-cadherin or claudin-5. In this revised version of the manuscript we assembled two new figures with new panels where we show, for each marker, ECs within the lesions and ECs in surrounding normal vessel as reference. These images have been inserted in Fig. 4 and Fig.5

We discuss in more detail the meaning of this cell reaction in Dejana and Lampugnani, Science 2019 and references within.

Rescue experiments should be carried out using CCM3 re-expression in CCM3^{-/-} ECs. Does this suppress EndMT marker genes or correct their (mal)functional properties?

Following the suggestion of the reviewer, we performed a rescue experiment re-expressing human CCM3 into immortalised *Ccm3* null ECs. After CCM3 re-expression the cells rescued the KO phenotype in terms of morphology, organization of adherence junctions and cell proliferation. Moreover, they partially downregulated the EndMT markers. The last point could be explained with the fact that the acquisition of mesenchymal phenotype is a long process, thus, the complete rescue would take longer time. These data have been inserted in the text at line 118 page 5, 247-253 page 11 and in Extended Data Fig. 1 and Extended Data Fig. 2.

What is the fate/function of CCM3^{-/-} ECs in vivo? The authors should re-inject these cells into syngeneic mice to assay their morphology and function (e.g. anastomosis/perfusion). Do they form aberrant vascular structures when re-injected? These experiments could be done using CCM3 rescue in CCM3^{-/-} ECs (see above).

Following the suggestion of the reviewer, we tested the capacity of *Ccm3* null ECs to generate cavernomas in a WT context. With this aim we isolated ECs from *Cdh5(PAC)-Cre-ER^{T2}/Ccm3^{fl}/R26R-Confetti* mice and re-injected into brains of WT adult mice. After 4 days, Confetti positive cells formed cavernomas, that also comprised WT non-labelled cells. This further confirmed the capacity of *Ccm3* null ECs to generate cavernomas recruiting WT ECs. Furthermore, to further test the idea of recruitment of WT cells in the cavernomas, we isolated ECs from non-labelled *Ccm3*-null mice and injected them into brains of *Ccm3*-WT, *Confetti* positive adult mice. We observed a significant incorporation of Confetti positive WT cells in the malformations. This further confirmed that *Ccm3*-null ECs are able to form abnormal vascular lesions also by recruiting WT cells. These new data have been inserted in the text at lines 200-213 page 9 and in Fig. 6. As suggested by this reviewer, we also tried to inject CCM3 rescued cells in wild-type brains, but we faced technical problems. In fact, freshly isolated cells resulted to be very sensitive to this particular viral infection; this resulted in high stress and low viability of the cells that could not be used for intracranial injection. On the other hand, injecting Polyoma Middle T immortalized cells raises problems since these cells tend to form angiomas independently from CCM deletion. (Williams R. et al, Cell 1989).

Since the authors suggest that CCM3^{-/-} ECs expand clonally in the CCM lesions, is it assumed that this expansion is via proliferation? If so, what is the link between CCM3 and cell cycle? Is deletion of CCM3 sufficient to drive S-phase or inhibit apoptosis in ECs? The authors should test this possibility experimentally.

As the reviewer has pointed out, we propose that, upon *Ccm3* deletion, ECs become more proliferative to form lesions. We have previously published that ECs in lesions were in a proliferative state as they

expressed ki67 (Bravi et al. PNAS 2015). It is also well documented (Lambertz et al. BMC Cancer 2015; Schleider et al. Neurogenetics 2011; Chen et al. Stroke 2009) that *Ccm3* plays an important role in regulating cell cycle in different cell types. To further test this aspect, as requested by the reviewer, in this paper we studied in even more detail the effect of *Ccm3* deletion on the cell cycle. These new data are discussed at lines 111-119 page 5 and more in detail in Extended Data Results and Extended Data Fig. 2. Briefly, we showed that the loss of *Ccm3* is sufficient to increase the proliferation rate of endothelial cells and to drive the entrance into the S-phase. These effects were inhibited by the re-constitution of *Ccm3* in *Ccm3* null cells.

The endothelial progenitor marker staining (CD157/CD200) is difficult to see and needs quantification. This staining also looks perivascular? Related to this, what fraction of Procr+ cells are labelled? Staining for Procr should be carried out. The concept that a small number of EC progenitors are the “drivers” of CCM lesions is compelling, but the data needs to be presented more clearly.

Cd157 and Cd200 stain cells positive for PECAM1, thus supporting the concept that they are endothelial cell progenitors. However, we agree that with the available antibodies the staining does not give an unequivocal response. We decided therefore to remove these stainings from the revised version of the manuscript. From a RNAseq analysis previously performed on *Procr*⁺ endothelial progenitors (Yu Q. et al. Cell Research 2016), *Procr*⁺ progenitors do not express higher levels of Cd157 and Cd200 if compared to *Procr*⁻ mature ECs. This fact suggests that the *Procr*⁺ and the *Cd157/Cd200* positive cells are two populations of progenitors that are independent, or that have only partial overlap. This is not surprising as distinct populations of stem/progenitor cells have been described by different groups in the last years, and that a hierarchy of progenitor cells has been proposed, as for instance for *Cd157*⁺ and *Cd200*⁺ cells in the work by Takakura's group.

Therefore, we thought to focus on the *Procr*⁺ progenitors as we report data that they are implicated in the formation of the cavernomas. To this aim, we performed a RNAscope in situ hybridization and showed that *Procr* is expressed in about 17% ECs lining cavernomas, with high variability within large cavernomas. Moreover, we showed by FACS analysis that *Procr*⁺ endothelial progenitors are 4.4 % of total ECs in wild type brain just after birth (P2), that is when we induced recombination to generate CCM lesions. This percentage is in line with what has been reported for other tissues (Yu Q. et al. Cell Research 2016). We also showed by RNAseq performed on WT cells from cerebellum at different ages that the expression of *Procr* declines over time. Moreover, by crossing the *Procr*^{CreERT2-IRES-tdTomato/+}/*Ccm3*^{fl/fl} mouse model with the R26-*EYFP* reporter we confirmed that the ECs in the cavernomas derived from *Procr*⁺ endothelial progenitors. Taken together these data strongly support the concept that a small number of *Procr*⁺ endothelial progenitors are able to trigger the formation of cavernomas in response to deletion of *Ccm3*. These data are now inserted in the text at lines 291, 295-299, 307-310 page 12-13 and in Fig. 8.

Can the authors assay the CCM gene product (PDCD10) and show mosaicism in human CCM lesions (PDCD10+ ECs and PDCD10- ECs)?

The mosaicism of mutation of ECs in human cavernomas has been described by Akers et al. Hum Mol Genet. 2009. In this study the authors using laser capture microscopy isolated ECs directly from the lesion and identified the mutation in only 15% of ECs. The authors concluded: “*This suggests that the somatic mutation occurs within a subset of the endothelial cells lining the caverns within this late-stage CCM lesion*”. As suggested by the reviewer, we attempted to perform CCM3 staining on human samples. We tested 5 commercially available antibodies from 4 different companies, but they did not work. Further trouble shooting would have been necessary but unfortunately sample from human CCM3 genetic cavernomas are very rare and we managed to get only few slides.

Do culture expanded CCM3-/- ECs retain or show enrichment for EC progenitor markers compared to their CCM3+/- counterparts?

Affymetrix analysis conducted on primary brain endothelial cells in culture revealed that *Ccm3* null cells do not express higher levels of progenitor's markers (*Procr*, *Peg3/PW1* and *Cd200*) except for a slight increase of *Cd157* (1.9 fold change).

The authors suggest that CCM3-/- ECs recruit/convert CCM3+/- ECs, but do CCM3+/- ECs block/revert/normalize CCM3-/- ECs in any way? This should be tested experimentally.

This is an interesting point raised by the reviewer. In the co-culture experiment we show that WT cells recruited into the spheroids upregulated some EndMT markers and downregulated Claudin5, when compared to normal WT cells. In the same experiments, we compared KO cells isolated from the spheroids with normal KO cells, cultured alone. As shown in Fig. 7, KO cells forming the spheroids express similar levels of EndMT markers than KO cells cultured alone.

This suggests that the co-culture did not influence the phenotype of the KO cells.

Moreover, in the wound healing experiment when WT cells get in contact with KO (Extended Data Fig. 8, and Movie 10-11) cells underwent changes in morphology and migratory behavior, but not *vice versa*. Taken together these data suggest that WT cells cannot revert/normalize the phenotype of KO cells. This means that the rescue of KO phenotype cannot be rescued by direct contact or soluble factors secreted by the WT cells, but necessarily needs the re-expression of CCM3, as described above.

Is there any evidence that CCM3^{+/+} ECs are recruited systemically (i.e. bone marrow)? Or are these locally-derived ECs from nearby vessels? A bone marrow origin is doubtful but maybe a point worthy of discussion if the authors have an opinion or data.

The evidence reported so far (Qing et al. Cell Research 2016) supports the idea that Procr⁺ cells are resident in mature vessels and do not come from the circulation. However, the experiment to final exclude the bone marrow origin of the cells forming the cavernoma would be a bone marrow transplantation after irradiation. Unfortunately, this procedure cannot be done on new born mice, while inducing the recombination at adult stage, after bone marrow transplantation, will not produce any CCM lesion. Thus, we cannot completely exclude a bone marrow origin of ECs in the lesions, even though all the evidence suggest that they come from resident progenitors.

We discussed this point in the text at lines 351-355 page 15.

Reviewer #4 (Remarks to the Author):

The work by Malinverno and colleagues builds upon previous findings by the Dejana laboratory suggesting that endothelial cells (ECs) lining cavernomas, the hallmark vascular lesions of the cerebrovascular disorder Cerebral Cavernous Malformations (CCM), are morphologically and molecularly different from ECs of the same vessel outside the lesion site. Familial forms of CCM are underlain by mutations in one of three genes (CCM1-3), and the two-hit mechanism (a germline mutation followed by a somatic mutation in a small number of ECs) has been employed to explain disease pathogenesis. In this manuscript, the authors test the hypothesis that CCM lesions originate from a small number of Ccm3-mutant ECs, which expand clonally, recruit surrounding wild type ECs, and induce their molecular and morphological transformation.

The manuscript addresses an interesting and novel idea, which is tested using acute and chronic mouse models of CCM3 disease, additional inducible transgenic mice, as well as in vitro co-culture assays of wild type or Ccm3 mutant ECs. The authors present a wealth of diverse and detailed data arguing in support of the proposed hypothesis. The study has considerable strengths, but also significant weaknesses. The major strength of the study is the novelty of the idea tested, namely, that clonal expansion of ECs mechanistically explains cavernoma formation. The major weakness, on the other hand, is that the conclusion that cerebral cavernomas arise from clonal expansion of ECs (as also stated in the title) is not fully supported by the data.

Below are the major points I wish to raise about the study:

1) Figure 1

The authors should include a picture of wild type retina, and not of (or, in addition to) wild type brain (current panel 1a). If the goal is to illustrate the clonal nature of retinal lesions, then the reference should be an image of the wild type retina. Is it possible that cavernomas have clonal origin because they arise from a vessel that is itself clonal?

As indicated by the reviewer, we included a picture of a confetti positive *Ccm3^{+/+}* retina as reference which do not show clonal regions (Fig. 1a). Moreover, clonal expansion in wild type resting retina from new born (P5) mice has been excluded by Manavski Y. et al. Circ Res 2018.

The lesions (magnification panel 1) in panels 1c and 1c' are shown as being representative of a "larger" lesion; however, they do not have the appearance of mature lesions with multiple caverns. Moreover, a section through a (any) lesion is not necessarily perpendicular to the wall of the dilated vessel (as it could be at an angle). Consequently, without a true 3D reconstruction, (and not just assessment of 10 thin sections [unclear if they are serial]), lesion area and diameter cannot be correctly assessed. This is an important point for the premise of the study, as small diameter lesions are said

to be clonal, whereas, in contrast, large cavernomas to be mosaic. This point becomes even more relevant as it is difficult to assess the non-clonality of the large lesions presented in panel 1d (too few cells of a different origin compared to the vast majority of the lesion that appears to be rather clonal).

In order to improve the quality of the images and make them more clear we have applied a Clarity protocol to brains and then performed 3D reconstruction of cavernomas. 3D reconstruction of lesions of different size and morphology, i.e. “small”, “large” and “mulberry” are shown in Fig. 1, Fig. 2, Fig. 3 and Extended Data Movie 1-8.

In particular, mulberry lesions with multiple caverns are shown in Fig. 1f and Extended Movie 6.

Regarding the last point raised by the reviewer about the non-clonality of large cavernomas, in our analysis we defined “clonal” lesions which showed a predominance of one colour, that is the case of most small lesions. In contrast, larger lesions are composed by more than one region of clonal expansion, as seen in Fig. 1f, as if adjacent clonal lesions fused together; as we discussed at lines 109-110 page 5. We are therefore tempted to define small lesions as monoclonal and large cavernomas as polyclonal; however, we cannot completely exclude that two contiguous cells got by chance the same colour after recombination and cooperate in the formation of a lesion, as also discussed at lines 141-157 page 6-7. For these reasons the definition “monoclonal” is not correct.

2) Extended Figure 1a

Please correct the schematic (it is indicated lethal at P10, however the text -line 98- states “lethal in around 12 days”

In the revised version of the manuscript we have re-shaped the figure and removed the scheme.

3) Figure 2d, e and Extended Figure 1d (text line 104)

The comparison attempted here is not valid – the acute model (P8) cannot be compared to the chronic model at P30. The comparison should be carried out within the same model.

The aim of this comparison was to characterise for the first time the novel chronic model and to compare it to the acute model. However, we agree that, in order to better describe the kinetic of lesions development the comparison has to be carried out in the same model. To solve this and the following issues, we performed a time course analysis of cavernomas at P8, p14 and P30. The characterization of the model at the different time points has been inserted in the text at lines 126-127, 133-134 page 6 and in Fig. 2

4) In order to conclude that cavernomas form via progression from a clonal, small/immature/early lesion to a non-clonal, larger/mature lesion (in which wild type cells are being recruited and induced to change in the presence of Ccm3-mutant cells), lesion progression (and accompanying molecular changes) should be documented in the chronic models. Specifically, staining with EMT markers over time will be necessary on smaller lesions that are 100% positive vs. larger (more mature) lesions, which should be mosaic. The authors performed a similar comparative analysis focusing on KLF4-positive cells (Fig. 2e), which should be extended to earlier time points in the same (chronic) model.

As required by the reviewer, in order to document lesion progression in the chronic model, we repeated the staining of EndMT markers at early, (P8) and late stage (P30). We assembled a new figure in which we show, for each considered EndMT gene, that:

- 1) most of the ECs lining small cavernomas show upregulation of the EndMT gene compared to the surrounding normal vessels;
- 2) in larger cavernomas few cells express the EndMT marker while the others are comparable to the surrounding normal vessels.

In parallel we also showed with the labelling for YFP, that while small cavernoma are mostly composed of YFP⁺, thus KO cells, larger cavernomas also comprise non-recombined, thus WT ECs. This data have been inserted in the text at lines 176-185 page 8, 189-190 same page, and in Fig. 4 and Fig. 5a.

5) In Figure 2f-m (presumably corresponding to the slow progression/chronic model – please clarify in the legend), the quality of the immunofluorescence could be improved. PECAM1 staining appears not to outline the lining of the lesions in its entirety; as a consequence, some, presumably, ECs appear positive for the mesenchymal markers, but lack PECAM1 staining. Can the authors show in the same model that smaller/immature/clonal lesions are 100% positive for EMT markers, whereas larger/mature lesions show a significant reduction of ECs that are positive for EMT markers?

We apologise since the images resulted unclear. In order to improve the quality of the images, we performed a new set of staining and assembled a new figure (Fig. 4), as also explained above, in which we showed that larger/mature lesions show a significant reduction of ECs positive for EndMT markers.

6) *Extended Figure 2c*

The in situ hybridization data are not convincing: this figure could be improved by DIC microscopy to visualize the outline of the cells; please comment on the staining outside the lesions.

Ccm3 is not endothelium specific but ubiquitously expressed. Therefore, staining outside the vessels is expected. Moreover, as the hybridization has been performed on low recombined mice, it is expected that ECs in normal vessels did not undergo recombination and thus express *Ccm3*. As requested we repeated the labeling and substituted the VinaGreen With DAB as chromogen in order to improve the quality of the images. The corresponding figure is now Extended Data Figure 3.

7) *Extended Figure 3*

Please specify that these are immortalized lung endothelial cells

We specified in the legend of the figure which in the revised manuscript is Extended Data Fig. 5

8) *Figure 4* (please note that in the revised manuscript this is Figure 7. A/N)

i) Panels b, d, and e: it is very difficult to assess the number of wild type ECs recruited in the Ccm3-mutant spheroids.

In order to assess the number of WT cells recruited, we counted the number of wild type and KO in the spheroids. Since the size of spheroids is very heterogenous, a more informative data is the percentage of cells of the two genotypes. We calculated that spheroids are composed by 26.76 ± 3.19 % of wild type cells. This new data have been inserted in the text at lines 220 page 9.

ii) Panel f and lines 174 and 189. The observed changes in migration rate in response to conditioned media from Ccm3-mutant EC cultures are mild and do not support the conclusion of the involvement of secreted factors (see also points iii and iv below).

The increase of migration rate in response to conditioned medium is milder than the one observed for instance upon stimulation with BMP6, but still around a 3-fold change and statistically significant. The milder response to conditioned medium could be due to its intrinsic undefined composition, enriched in secreted factors but possibly poor in nutrients. However, the treatment with the BMPs inhibitor DMH1 completely abolish the increase of migration rate, and we think that this data supports a partial role of secreted factors. We believe anyway that secreted factors cooperate with direct cell-cell contact as discussed above at point iv.

iii) Panel f: There is a discrepancy between the data on the rate of migration described in this panel (3 um/hr with conditioned media from wild type cultures vs. 8 um/hr with conditioned media from Ccm3-mutant cultures) and data in Extended Figure 7e (7 um/hr with vehicle as negative control). How do the authors explain this?

The negative control in Extended Figure 7e (now Extended Figure 10) is fresh medium, while in the Figure 4f (now Figure 7f) the control is conditioned media from wild type culture. The conditioned media are enriched in secreted factors, but can also be depleted of nutrients, thus resulting in a slowdown in migration rate.

iv) Line 162 and Extended Figure 6. As the authors mention, involvement of cell-cell contacts in mediating the recruitment of wild type cells by Ccm3-mutant ECs is indeed plausible.

We believe that cell-cell contact is involved in the recruitment of wild type cells, as also described in Extended Data Fig. 8. Nevertheless, we think that cell-cell contact and secreted factors are not in contrast or mutually exclusive, but they may cooperate to mediate the recruitment of wild type cells by *Ccm3*-mutant ECs. We therefore discussed more in detail in the text at line 261 page 11.

v) Panel g and lines 165-168): the comparison here is between wild type cells isolated from the spheroids vs. wild type cells cultured as a monolayer. This is problematic, and a more accurate comparison would have been with wild type cells

that have not been recruited/encapsulated in the spheroids, yet cultured in the same dish. An additional problem is that by definition cells that grow in 3D (such as within a spheroid), undergo morphological (and molecular) changes that distinguish them from the same cells grown as a monolayer.

As suggested by the reviewer we compared the gene expression profile of WT cells recruited into the spheroids with WT cells cultured in the same dish and not recruited, as now indicated in Fig. 7g and in the text at lines 235-236 page 10.

In order to exclude that the molecular changes which underwent the recruited cells were not induced only by the 3D culture, we performed a spheroid assay with WT or KO cells kept separated. WT cells cultured as spheroids showed only minimal increase of some EndMT markers not comparable with the modifications induced by the co-culture. Therefore, we can assess that the molecular modifications to which underwent WT cells recruited are induced by the presence of KO cells rather than by the 3D morphology.

These new data have been inserted on the text at lines 236-244 page 10 and in the Extended Data Fig. 9

9) line 170: how do the authors explain the increase on Fsp1 relative expression only in the presence of BMP6, but not upon exposure to conditioned media from Ccm3-mutant cultures?

The treatment with BMP6 has been done in starvation medium with high concentration of the agonist, that led to an activation of the BMP/SMADs pathway and upregulation of EndMT markers. On the contrary, the conditioned medium has a non-defined composition and can contain several soluble factors. As shown in Extended Data Fig. 10a, BMP6 is one of them, but we do not exclude that other secreted factors can cooperate in modulating, even negatively, the EndMT. Moreover, the heterogeneity of response of cells to different stimuli reflects the complex nature of the mesenchymal transition, that is not a step-by-step process, but is a continuum of phenotypical modifications which occur with different kinetics.

10) Although the source of the different materials used in every figure/panel is described in Methods, having to constantly refer to the Methods to retrieve this information is disruptive. It would be preferable if this information is also provided as needed throughout the text and in the figure legends (e.g. ECs in culture are either primary brains ECs or lung ECs; FACS is performed not on mice, but specifically on brains, etc.)

As suggested by the reviewer, in order to make the reading easier we provided more information regarding the Methods used directly in the legends.

In summary, the work presented in this manuscript is intriguing and proposes a novel mechanism to explain the formation of CCM lesions. However, additional experiments are needed to support the conclusions reached by the authors, as well as the definitive nature of the title-statement, which in this reviewer's opinion is very strongly worded, without being fully substantiated by the findings as these are currently detailed, and thus should be revised.

REVIEWERS' COMMENTS:

Reviewer #1 (Remarks to the Author):

The revised manuscript is improved and my concerns were addressed. This is an important contribution to the field.

Reviewer #3 (Remarks to the Author):

The authors have satisfied all of the comments from this reviewer and the work is highly suitable for publication in Nature Comms.

Reviewer #4 (Remarks to the Author):

The authors have addressed all but one of the points raised in my review. I continue to find the in situ hybridization data presented in Extended Data Figure 3 not convincing.

We would like to thank one last time the reviewers for their positive comments, which are reported below.

In the light of the last concern of reviewer #4, we agreed to remove the image of the in situ hybridization from Supplementary Figure 3. We believe nevertheless that all the other data in the paper support the concept that lesions are mosaic for CCM3 expression.

REVIEWERS' COMMENTS:

Reviewer #1 (Remarks to the Author):

The revised manuscript is improved and my concerns were addressed. This is an important contribution to the field.

Reviewer #3 (Remarks to the Author):

The authors have satisfied all of the comments from this reviewer and the work is highly suitable for publication in Nature Comms.

Reviewer #4 (Remarks to the Author):

The authors have addressed all but one of the points raised in my review. I continue to find the in situ hybridization data presented in Extended Data Figure 3 not convincing.